# Efficient Multi-Task Reinforcement Learning with Cross-Task Policy Guidance

**Jinmin He**[1,2], **Kai Li**[1,2,*] **Yifan Zang**[1,2], **Haobo Fu**[5], **Qiang Fu**[5], **Junliang Xing**[4,*] **Jian Cheng**[1,3]

[1]Institute of Automation, Chinese Academy of Sciences
[2]School of Artificial Intelligence, University of Chinese Academy of Sciences
[3]AiRiA      [4]Tsinghua University      [5]Tencent AI Lab
{hejinmin2021,kai.li,zangyifan2019,jian.cheng}@ia.ac.cn,
{haobofu,leonfu}@tencent.com, jlxing@tsinghua.edu.cn

## Abstract

Multi-task reinforcement learning endeavors to efficiently leverage shared information across various tasks, facilitating the simultaneous learning of multiple tasks. Existing approaches primarily focus on parameter sharing with carefully designed network structures or tailored optimization procedures. However, they overlook a direct and complementary way to exploit cross-task similarities: the control policies of tasks already proficient in some skills can provide explicit guidance for unmastered tasks to accelerate skills acquisition. To this end, we present a novel framework called Cross-Task Policy Guidance (CTPG), which trains a guide policy for each task to select the behavior policy interacting with the environment from all tasks' control policies, generating better training trajectories. In addition, we propose two gating mechanisms to improve the learning efficiency of CTPG: one gate filters out control policies that are not beneficial for guidance, while the other gate blocks tasks that do not necessitate guidance. CTPG is a general framework adaptable to existing parameter sharing approaches. Empirical evaluations demonstrate that incorporating CTPG with these approaches significantly enhances performance in manipulation and locomotion benchmarks.

## 1 Introduction

Deep reinforcement learning (RL) has undergone remarkable progress over the past decades, showcasing its efficacy across various domains, such as game playing [16, 28] and robotic control [12, 13]. However, most of these deep RL methods primarily focus on learning different tasks in isolation, making it challenging to utilize shared information between tasks to develop a generalized policy. Multi-task reinforcement learning (MTRL) aims to master a set of RL tasks effectively. By leveraging the potential information sharing among different tasks, joint multi-task learning typically exhibits higher sample efficiency than training each task individually [27].

A significant challenge in MTRL lies in determining what information should be shared and how to share it effectively. Recent studies have proposed various approaches to tackle this challenge via network parameter sharing with carefully designed network structures [10, 22, 27] or tailored optimization procedures [4, 14, 30]. We summarize such methods as *implicit knowledge sharing* in Section 2. Despite these unremitting efforts, another largely overlooked way exists to exploit cross-task similarities to improve the learning efficiency of multiple tasks. Intuitively, humans can effortlessly discern which skills can be shared from other tasks while learning a specific task. For instance, someone who can ride a bicycle can quickly learn to ride a motorcycle by referring to

---

*Corresponding authors.



(a) Button-Press v.s. Drawer-Close          (b) Door-Open v.s. Drawer-Open

Figure 1: Full or partial policy sharing in the manipulation environment. (a): Task *Button-Press* and *Drawer-Open* share almost the same policy, where the robotic arm needs to reach a specified position (button or handle) and then push the target object. (b): Task *Door-Open* and *Drawer-Open* share the policy of grabbing the handle in the first phase, but they are required to open the target object by different movements (rotation or translation).

related skills, such as operating controls, maintaining balance, and executing turns. Likewise, a motorcyclist adept in these skills can also quickly learn to ride a bicycle. This ability allows humans to efficiently master multiple tasks by selectively referring to skills previously learned. As shown in Figure 1, similar full or partial policy sharing is also evident in robotic arm manipulation tasks. These cross-task similarities enable *policy guidance*, *i.e.*, control policies of tasks already proficient in specific skills can generate valuable training data for unmastered tasks. Compared to the common practice, which blindly generates training trajectories for each task solely with its own control policy, generating training trajectories using a control policy from other tasks that perform better in the current situation can better facilitate the learning procedure. Moreover, this *explicit policy sharing* approach significantly reduces unnecessary exploration of similar contexts in different tasks.

The key challenge encountered in this approach to MTRL is discerning beneficial sharing control policies for each task adaptively. To address this challenge, [32] uses a Q-filter to identify single-step shareable behaviors without ensuring optimality for long-term policy sharing. In contrast, we propose a simple yet effective framework called Cross-Task Policy Guidance (CTPG) for more robust long-term policy guidance. Initially, we group the control policies of all tasks into a candidate set. Subsequently, for each task, we train a guide policy to identify useful sharing control policies, and then the chosen control policy generates better training trajectories to achieve *policy guidance*. Furthermore, we design two gating mechanisms to avoid unfavorable policy guidance interfering with learning. The first, policy-filter gate, leverages the value function to refine the candidate set by masking out control policies that are not beneficial for guidance. The second, guide-block gate, withholds extra guidance for the mastered easy tasks, allowing the focus to be on further solidifying the skills already acquired. With the incorporation of the above two gates, CTPG greatly improves the quality of the policy guidance, thereby fostering enhanced exploration and learning efficiency.

CTPG is a generalized MTRL framework that can be combined with various existing parameter sharing methods. Among these, we choose several classical approaches and integrate them with CTPG, achieving significant improvement in sample efficiency and final performance on both manipulation [31] and locomotion [11] MTRL benchmarks. Furthermore, we conduct detailed ablation studies to gain insights into how each component of CTPG contributes to its final performance.

## 2   Related Work

Multi-task learning is a training paradigm that enhances generalization by leveraging the information inherent in potentially related tasks [3, 19, 34]. Multi-task reinforcement learning extends this concept to reinforcement learning, expecting that information shared across tasks will be uncovered by simultaneously learning multiple RL tasks [25]. In this study, we distinguish between information sharing as implicit knowledge sharing and explicit policy sharing.

**Implicit Knowledge Sharing.**   Implicit knowledge sharing primarily focuses on sharing parameters or representations, but it encounters the challenge of negative knowledge transfer due to simultaneous updates within the same network. [14, 30] regard MTRL as a multi-objective optimization problem aimed at managing conflicting gradients resulting from different task losses during training. [22, 27]

partition the network into distinct modules and combine these modules to form different sub-policies for different tasks. [5, 21] endeavor to choose or learn better representations as more effective task-conditioned information for policy training. [7, 23] employ distillation and regularization to fuse separate task-specific policies into a unified policy for diverse tasks.

**Explicit Policy Sharing.**   Explicit policy sharing is expressed as the direct sharing of behaviors or policies between different tasks. [20] employs a hierarchical policy that decides when to directly use a previously learned policy and when to acquire a new one. Nonetheless, instead of learning multiple tasks simultaneously, it adopts a sequential task-learning approach, necessitating a manually well-defined curriculum of tasks. [32] uses a Q-filter to identify shareable behaviors. During exploration, each control policy proposes a candidate action, and the policy that suggests the maximum Q-value action on the source task is executed for the following timesteps. However, maximizing Q-value in a single timestep does not guarantee the optimality of this policy across continuous timesteps. CTPG is a new explicit policy sharing method that learns a guide policy for long-term policy guidance.

## 3   Preliminaries

**Multi-Task Reinforcement Learning.**   We aim to simultaneously learn $N$ tasks, where each task $i \in \mathbb{T}$ is represented as a Markov decision process (MDP) [1, 18]. Each MDP is defined by the tuple $\langle S, A, P_i, R_i, \gamma \rangle$, where $S$ denotes the state space, $A$ the action space, $P_i : S \times A \to S$ the environment transition function, $R_i : S \times A \to \mathbb{R}$ the reward function, and $\gamma \in [0, 1)$ the discount factor. In the scope of this work, different tasks share the same state and action spaces, distinguished by different transition and reward functions. The goal of the MTRL agent is to maximize the average expected return across all tasks, which are uniformly sampled during training.

**Soft Actor-Critic.**   In this work, we use the Soft Actor-Critic (SAC) [9] algorithm, an off-policy actor-critic method under the maximum entropy framework. The critic network $Q_\theta(s_t, a_t)$ parameterized by $\theta$, representing a soft Q-function [8], aims to minimize the soft Bellman residual:

$$J_Q(\theta) = \mathbb{E}_{(s_t, a_t, r_t) \sim \mathcal{D}} \left[ \frac{1}{2} \left( Q_\theta(s_t, a_t) - \left( r_t + \gamma \mathbb{E}_{s_{t+1} \sim P} \left[ V_{\bar{\theta}}(s_{t+1}) \right] \right) \right)^2 \right], \tag{1}$$

$$V_{\bar{\theta}}(s_t) = \mathbb{E}_{a_t \sim \pi_\phi} \left[ Q_{\bar{\theta}}(s_t, a_t) - \alpha \log \pi_\phi(a_t|s_t) \right], \tag{2}$$

where $\mathcal{D}$ represents the data in the replay buffer, and $\bar{\theta}$ is the target critic network parameter. The actor network $\pi_\phi(a_t|s_t)$ is parameterized by $\phi$, and the objective of policy optimization is:

$$J_\pi(\phi) = \mathbb{E}_{s_t \sim \mathcal{D}} \left[ \mathbb{E}_{a_t \sim \pi_\phi} \left[ \alpha \log \pi_\phi(a_t|s_t) - Q_\theta(s_t, a_t) \right] \right], \tag{3}$$

where $\alpha$ is a learnable temperature parameter to penalize entropy as follows:

$$J(\alpha) = \mathbb{E}_{a_t \sim \pi_\phi} \left[ -\alpha \log \pi_\phi(a_t|s_t) - \alpha \bar{\mathcal{H}} \right], \tag{4}$$

where $\bar{\mathcal{H}}$ is a desired minimum expected entropy. If the optimization leads to an increase in $\pi_\phi(a_t|s_t)$ with a decrease in the entropy, the temperature $\alpha$ will accordingly increase. In the following sections, we use subscripts to signify the networks specific to each task. Specifically, the control policy of task $i$ is represented as $\pi_i$, and the corresponding Q-value function is denoted as $Q_i$.

## 4   Cross-Task Policy Guidance

Explicit policy sharing offers a direct and efficient way to master multiple tasks. If a task is already mastered, its control policy can be fully or partially shared with other tasks to guide tasks requiring similar skills to be quickly learned. Instead of each task generating trajectories constantly by its corresponding control policy, as in most existing MTRL algorithms, we consider using control policies of other tasks to generate training data for the current task when appropriate. To achieve this goal, we propose a novel framework called Cross-Task Policy Guidance (CTPG), which extra learns a guide policy for each task to identify beneficial policies for guidance. We illustrate the trajectory generation process of Task 1 in Figure 2. For this task, its guide policy $\Pi_1^g$ selects a policy $\pi'$ from the candidate set of all control policies $\{\pi_i\}_{i=1}^N$ every fixed $K$ timesteps. It then uses $\pi'$ as the behavior policy to interact with the environment and collect data for the next $K$ timesteps.

The CTPG framework alters only the data collection process, guiding the control policy training through better exploration trajectories. In Section 4.1, we introduce the guide policy in detail and propose a hindsight off-policy correction mechanism for its training. In addition, we propose two gating mechanisms to enhance the efficiency of CTPG: the policy-filter gate discussed in Section 4.2 and the guide-block gate detailed in Section 4.3.

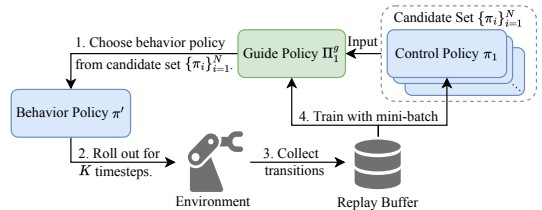

Figure 2: Overview of the CTPG framework.

## 4.1 Guide Policy

The control policy $\pi_i(a_t|s_t)$ of task $i$ maps the state $s_t$ to the environment action $a_t$, and its corresponding Q-value function $Q_i(s_t, a_t)$ estimates the expected return. The guide policy $\Pi_i^g(j_t|s_t)$ of task $i$ outputs a *task index* $j_t \in \mathbb{T}$, and the corresponding control policy $\pi_{j_t}$ of task $j_t$ serves as the behavior policy. The guide Q-value function $Q_i^g(s_t, j_t)$ estimates the expected return of using $\Pi_i^g$ to select different control policies for every $K$ timesteps, with its Bellman equation defined as follows:

$$\mathcal{B}^{\Pi_i^g} Q_i^g(s_t, j_t) \triangleq R_i^g(s_t, j_t) + \gamma^K \mathbb{E}_{j_{t+K} \sim \Pi_i^g, s_{t+K} \sim P_i} \left[ Q_i^g(s_{t+K}, j_{t+K}) \right], \tag{5}$$

where $\mathcal{B}^{\Pi_i^g}$ is the Bellman operator of the guide policy $\Pi_i^g$, and the corresponding reward function $R_i^g$ is defined as the expected cumulative discount rewards for behavior policy $\pi_{j_t}$ over $K$ timesteps:

$$R_i^g(s_t, j_t) = \mathbb{E}_{a_{t'} \sim \pi_{j_t}, s_{t'+1} \sim P_i} \left[ \sum_{t'=t}^{t+K-1} \gamma^{t'-t} R_i(s_{t'}, a_{t'}) \right]. \tag{6}$$

For each task $i$, during trajectory generation, CTPG first utilizes its guide policy $\Pi_i^g$ to sample a behavior policy $\pi_{j_t}$ from the candidate set of all tasks' control policies:

$$j_t \sim \Pi_i^g(\cdot|s_t), \tag{7}$$

and then samples actions using the behavior policy $\pi_{j_t}$ for the next $K$ timesteps:

$$a_{t'} \sim \pi_{j_t}(\cdot|s_{t'}), \tag{8}$$

where $t' \in \{t, t+1, \ldots, t+K-1\}$. After each timestep, we obtain the reward $r_{t'} = R_i(s_{t'}, a_{t'})$ and the next state $s_{t'+1}$. The transition $\langle i, s_{t'}, a_{t'}, j_t, r_{t'}, s_{t'+1} \rangle$ is stored in the replay buffer.

The guide policy is trained to maximize the expected return of the current task by choosing appropriate control policies in certain states. If the control policies of some tasks already proficient in specific skills can be shared with the current task in similar states, the guide policy can quickly learn to use these control policies in those states to generate better training data for the current task. The guide policy and the control policy are trained simultaneously. We can use any off-policy RL algorithms for control policy training and any on/off-policy RL algorithms for guide policy training. In this work, we use SAC [9] for the control policy training and the discrete action space variant of SAC [6] for the guide policy training. Given that the guide policy acts every $K$ timesteps, its training frequency is $1/K$ that of the control policy. The detailed pseudo-codes for the control policy and guide policy training are provided in Appendix A.1 (Algorithms 1 and 2).

**Hindsight Off-Policy Correction.** During off-policy training, the guide policy faces a non-stationary challenge. Since the control policies are continually updated during the training of the guide policies, the actions chosen by the behavior policies during data collection may no longer align with the improved corresponding control policies, thereby compromising the validity of the training experience. We address this concern by implementing a hindsight off-policy correction mechanism that reassigns the action $j_t$ sampled by the past guide policy to a new one $j_t'$, whose control policy $\pi_{j_t'}$ is more likely to output the historical action sequence $\{a_{t'}\}_{t'=t}^{t+K-1}$. Specifically, we utilize maximum likelihood estimation following:

$$j_t' = \arg\max_j \prod_{t'=t}^{t+K-1} \pi_j(a_{t'}|s_{t'}) = \arg\max_j \sum_{t'=t}^{t+K-1} \log \pi_j(a_{t'}|s_{t'}). \tag{9}$$

In this way, we can leverage past experiences effectively to train the guide policy. The workings of the hindsight off-policy correction mechanism in SAC are detailed in Appendix C.

## 4.2 Not All Policies Are Beneficial for Guidance

In Section 4.1, we set the guide policy's action space as the set of all control policies $\{\pi_i\}_{i=1}^N$. However, not all control policies within the action space of $\Pi_i^g$ are beneficial for task $i$ in state $s_t$. Some control policies perform even worse than the current task's own control policy $\pi_i$, rendering them ineffective for guidance. To address this issue, we design a policy-filter gate to refine the action space of the guide policy by adaptively filtering out unfavorable control policies in state $s_t$. The trajectory generation process solely using the current task's control policy $\pi_i$ can be regarded as equipped with a special guide policy $\Pi_i^{\tilde{g}}$ that exclusively selects $\pi_i$ as the behavior policy, *i.e.*, $\Pi_i^{\tilde{g}}(i|s_t) = 1$ for any $s_t$. The guide Q-value $Q_i^{\tilde{g}}$ of $\Pi_i^{\tilde{g}}$, defined by Equation 5, is:

$$
\begin{aligned}
Q_i^{\tilde{g}}(s_t, i) &= R_i^g(s_t, i) + \gamma^K \mathbb{E}_{s_{t+K} \sim P_i}\left[ Q_i^{\tilde{g}}(s_{t+K}, i) \right] \\
&= \mathbb{E}_{a_{t'} \sim \pi_i, s_{t'+1} \sim P_i}\left[ \sum_{t'=t}^{t+K-1} \gamma^{t'-t} R_i(s_{t'}, a_{t'}) + \gamma^K Q_i^{\tilde{g}}(s_{t+K}, i) \right] \\
&= \cdots \\
&= \mathbb{E}_{a_{t'} \sim \pi_i, s_{t'+1} \sim P_i}\left[ \sum_{t'=t}^{\infty} \gamma^{t'-t} R_i(s_{t'}, a_{t'}) \right] \\
&= V_i(s_t),
\end{aligned}
\tag{10}
$$

where we repeatedly expand $Q_i^{\tilde{g}}$ to find that $Q_i^{\tilde{g}}(s_t, i)$ is equal to $V_i(s_t)$, the state value function of task $i$'s control policy. Because the value function can serve as a filter for high-quality training data [17, 29, 33], it becomes intuitive to judge the quality of the behavior policy $\pi_{j_t}$ following guide policy and the current task's control policy $\pi_i$ by directly comparing $Q_i^g(s_t, j_t)$ and $V_i(s_t)$. In our implementation, we estimate $V_i(s_t)$ via Monte Carlo sampling of $Q_i(s_t, a_t)$ with $a_t \sim \pi_i(a_t|s_t)$ [15]. Relying on this mechanism, the policy-filter gate serves as a mask vector $m(s_t)$ to indicate whether each control policy is beneficial for guidance in state $s_t$. Specifically, each element of $m(s_t)$ is:

$$
m_j(s_t) = \begin{cases} 1, & Q_i^g(s_t, j) \geq V_i(s_t), \\ 0, & Q_i^g(s_t, j) < V_i(s_t), \end{cases} \quad \text{for } j \in \{1, 2, \ldots, N\},
\tag{11}
$$

where $j$ indicates the element index and task index. Then, the behavior policy $\pi_{j_t}$ is sampled by:

$$
j_t \sim \text{Normalize}\left(\Pi_i^g(\cdot|s_t) \cdot m(s_t)\right).
\tag{12}
$$

If none of the control policies are beneficial for guidance, *i.e.*, $m(s_t) = \mathbf{0}$, it indicates that the current task's control policy $\pi_i$ is the most proficient within the current state, rendering other control policies unnecessary for enhancing trajectory generation.

**Comparable Guide Q-Value.** Typically, RL algorithms estimate Q-values to approximate the expected return of the current state-action pair, allowing for the calculation of the policy-filter gate in Equation 11 through a direct comparison of $V_i(s_t)$ and $Q_i^g(s_t, j_t)$. However, in maximum entropy RL algorithms such as SAC, Q-value estimation incorporates the maximum entropy objective, leading to the incomparability of two policies with different entropy objectives. Therefore, we learn another comparable guide Q-value $\hat{Q}_i^g$ with discounted entropy of $\pi_i$ following:

$$
\begin{aligned}
\hat{Q}_i^g(s_t, j_t) &= \mathbb{E}_{a_{t'} \sim \pi_{j_t}, s_{t'+1} \sim P_i}\left[ \sum_{t'=t}^{t+K-1} \gamma^{t'-t} \left( R_i(s_{t'}, a_{t'}) + \alpha_i \mathcal{H}(\pi_i(\cdot|s_{t'})) \right) \right] \\
&\quad + \gamma^K \mathbb{E}_{j_{t+K} \sim \Pi_i^g, s_{t+K} \sim P_i}\left[ \hat{Q}_i^g(s_{t+K}, j_{t+K}^g) \right].
\end{aligned}
\tag{13}
$$

Since both $\hat{Q}_i^g(s_t, j_t)$ and $V_i(s_t)$ estimate the return with the entropy of the current task's control policy, they can be directly compared to assess whether control policies are beneficial for guidance. A detailed comparability analysis of this comparable guide Q-value in SAC is provided in Appendix B.

## 4.3 Not All Tasks Need Guidance

When simultaneously learning multiple tasks, easy tasks converge faster than difficult ones. The control policies of easy tasks allow for the quick acquisition of some effective skills, which may

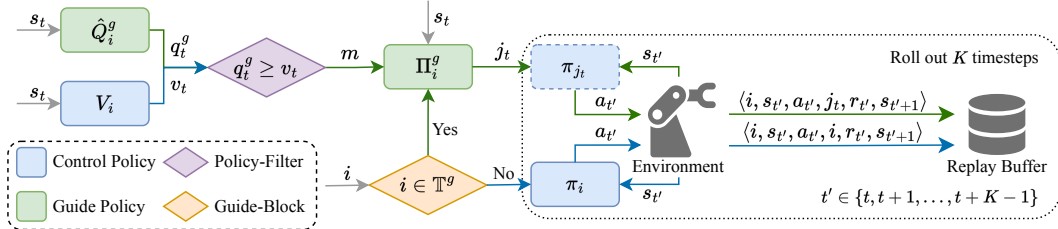

Figure 3: Illustration of the comprehensive CTPG framework. Initially, the guide-block gate selectively provides guidance on tasks $i \in \mathbb{T}^g$. Subsequently, the policy-filter gate generates a mask $m$ to sift through the beneficial policies. Finally, the policy chosen by the guide policy or the control policy of the current task itself interacts with the environment over $K$ timesteps to collect training data.

be helpful in exploring other tasks. However, these mastered or easy tasks do not need additional guidance from other policies; instead, they focus on further solidifying their already acquired skills. Therefore, not all tasks require assistance from the guide policy.

Based on the above analysis, we design another guide-block gate to prevent the guide policy from engaging in tasks that do not necessitate guidance. This mechanism is directly related to SAC's temperature coefficient $\alpha_i$. For difficult tasks $i_{\text{diff}}$, their control policy entropies $\mathcal{H}\left(\pi_{i_{\text{diff}}}(\cdot|s_t)\right)$ tend to be high, and the corresponding temperature parameters $\alpha_{i_{\text{diff}}}$ decrease according to Equation 4. Conversely, the temperature parameters $\alpha_{i_{\text{easy}}}$ increase for easy tasks $i_{\text{easy}}$. Therefore, $\alpha_i$ is a metric reflecting the relative difficulty and mastery of different tasks. We form the tasks that require guidance into a subset $\mathbb{T}^g$ following:

$$\mathbb{T}^g = \left\{ i \mid \log \alpha_i \leq \frac{1}{N} \sum_{j=1}^{N} \log \alpha_j \right\}, \tag{14}$$

which selects the tasks by comparing the difficulty of the current task versus the average of all tasks. For tasks $i \notin \mathbb{T}^g$, the guide-block gate restricts them from using the guide policy, so they only use their own control policies to interact with the environment. Consequently, the guide policy can stop training with samples from the tasks $i \notin \mathbb{T}^g$ and focus on learning the guidance for unmastered tasks.

The comprehensive CTPG framework, with two special gating mechanisms, is summarized in Figure 3. The complete pseudo-code for CTPG is described in Appendix A.2 (Algorithm 3).

## 5 Experiments

The experiments are designed to answer the following research questions: **Q1:** Can implicit knowledge sharing approaches be combined with CTPG to further improve performance? **Q2:** Does the guide policy in CTPG learn useful sharing policies? **Q3:** How does each component within CTPG contribute to the final performance? **Q4:** How does CTPG perform without implicit knowledge sharing approaches? **Q5:** Can CTPG expedite the exploration of new tasks effectively?

### 5.1 Environments

We conduct experiments on MetaWorld manipulation and HalfCheetah locomotion MTRL benchmarks, selecting two setups for evaluation within each benchmark.

**MetaWorld Manipulation Benchmark.** The MetaWorld benchmark [31] consists of 50 robotics manipulation tasks employing a sawyer arm in the MuJoCo environment [24]. It provides two setups: *MetaWorld-MT10*, comprising a suite of 10 tasks, and *MetaWorld-MT50*, comprising a suite of 50 tasks. Following the settings in [9], the goal position is randomly reset at the start of every episode. We use the mean success rate as our evaluation metric, which is clearly defined in the environment.

**HalfCheetah Locomotion Benchmark.** The HalfCheetah is a 6-DoF walking robot consisting of 9 links and 8 joints connecting them in the MuJoCo environment [24]. The multi-task benchmark HalfCheetah Task Group [11] contains different HalfCheetah robots. *HalfCheetah-MT5* includes 5

Table 1: Quantitative result of five classical implicit knowledge sharing approaches combined with different explicit policy sharing methods. The two HalfCheetah locomotion environments are measured on episode return, and the two MetaWorld manipulation environments are measured on success rate. We highlight the best-performing explicit policy sharing method in bold and annotate the best combination of two information sharing methods with boxes.

| Environment | Method | MTSAC | MHSAC | PCGrad | SM | PaCo |
|---|---|---|---|---|---|---|
| HalfCheetah MT5 (× 1e3) | Base | $9.16 \pm 0.42$ | $8.68 \pm 0.55$ | $9.57 \pm 0.73$ | $9.57 \pm 0.21$ | $7.18 \pm 0.44$ |
| | *w/* QMP | $8.81 \pm 0.22$ | $9.09 \pm 0.64$ | $9.46 \pm 0.57$ | $10.09 \pm 0.53$ | $7.83 \pm 0.28$ |
| | *w/* CTPG | $\mathbf{9.59 \pm 0.40}$ | $\mathbf{9.25 \pm 0.12}$ | $\mathbf{10.27 \pm 0.40}$ | $\boxed{\mathbf{10.47 \pm 0.34}}$ | $\mathbf{7.95 \pm 0.47}$ |
| HalfCheetah MT8 (× 1e3) | Base | $9.00 \pm 0.88$ | $8.90 \pm 0.60$ | $10.17 \pm 1.06$ | $10.05 \pm 0.55$ | $8.44 \pm 0.56$ |
| | *w/* QMP | $10.00 \pm 0.47$ | $9.61 \pm 0.54$ | $10.65 \pm 0.43$ | $10.41 \pm 0.61$ | $\mathbf{9.28 \pm 0.48}$ |
| | *w/* CTPG | $\mathbf{10.17 \pm 0.31}$ | $\mathbf{9.82 \pm 0.40}$ | $\boxed{\mathbf{11.09 \pm 0.50}}$ | $\mathbf{10.81 \pm 0.51}$ | $9.02 \pm 0.48$ |
| MetaWorld MT10 (%) | Base | $62.72 \pm 6.19$ | $63.51 \pm 2.97$ | $69.62 \pm 4.04$ | $74.52 \pm 2.29$ | $69.77 \pm 7.28$ |
| | *w/* QMP | $64.91 \pm 8.82$ | $65.87 \pm 3.05$ | $67.53 \pm 2.93$ | $69.78 \pm 7.50$ | $69.84 \pm 3.49$ |
| | *w/* CTPG | $\mathbf{75.76 \pm 3.82}$ | $\mathbf{74.94 \pm 2.97}$ | $\mathbf{73.31 \pm 3.66}$ | $\boxed{\mathbf{78.97 \pm 2.41}}$ | $\mathbf{70.40 \pm 3.62}$ |
| MetaWorld MT50 (%) | Base | $47.51 \pm 1.95$ | $52.04 \pm 2.78$ | $52.85 \pm 4.12$ | $55.04 \pm 2.84$ | $59.46 \pm 5.14$ |
| | *w/* QMP | $47.82 \pm 1.62$ | $51.79 \pm 4.83$ | $54.05 \pm 1.39$ | $55.91 \pm 5.08$ | $53.81 \pm 2.00$ |
| | *w/* CTPG | $\mathbf{55.97 \pm 2.56}$ | $\mathbf{56.91 \pm 2.57}$ | $\mathbf{58.91 \pm 2.10}$ | $\mathbf{66.24 \pm 3.37}$ | $\boxed{\mathbf{68.10 \pm 3.44}}$ |

tasks under various scales of simulated earth-like gravity, ranging from one-half to one-and-a-half of the normal gravity level. *HalfCheetah-MT8* includes 8 tasks with various morphology of a specific robot body part. We use the episode return as our evaluation metric.

Further information regarding environmental setups is provided in Appendix D.

## 5.2 Performance Improvement on Implicit Knowledge Sharing Approaches

CTPG is a generalized MTRL framework adaptable to various implicit knowledge sharing approaches, wherein both the control policy and the guide policy use a unified network. Specifically, in our implementation, the unified control policy employs the same network structure and update procedure as the implicit knowledge sharing approaches, and the unified guide policy utilizes a straightforward multi-head structure for parameter sharing.

To answer **Q1**, we choose five classical implicit knowledge sharing approaches: (1) **MTSAC** extends SAC for MTRL by employing one-hot encoding for task representation. (2) **MHSAC** utilizes a shared network backbone apart from independent heads for each task. (3) **PCGrad** [30] resolves issues arising from conflicting gradients among tasks through gradient manipulation. (4) **SM** [27] trains a routing network to propose weights for soft module combinations. (5) **PaCo** [22] learns a compositional policy where task-shared parameters combine with task-specific parameters to form task policies. Combined with these implicit knowledge sharing approaches, we compare CTPG against: (i) **Base** represents the source version of these approaches. (ii) **QMP** [32] employs a one-step Q-value filter to identify shareable behaviors.

We train all combinations with 0.8 million samples per task in the HalfCheetah locomotion benchmark and 1.5 million samples per task in the MetaWorld manipulation benchmark. Each combination is trained 5 times with different seeds. We evaluate the final policy over 100 episodes per task and report the mean performance and standard deviation across different seeds in Table 1.

Based on the experimental findings, it becomes evident that, except for the PaCo algorithm in HalfCheetah-MT8, the combination with CTPG leads to a notable enhancement in performance across all scenarios. QMP does not perform as well as CTPG because QMP only offers single-step behavior guidance, whereas CTPG's guide policy learns long-term policy guidance. Notably, as the number of tasks increases, CTPG exhibits superior performance due to the higher probability of explicit policy sharing between tasks, facilitating more effective policy exploration. Besides the final performance evaluation, comprehensive training curves are presented in Appendix E.1.

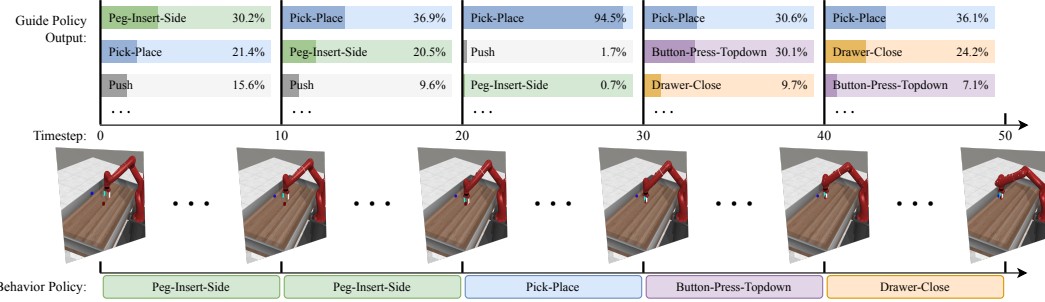

Figure 4: We display the state of task *Pick-Place* at every 10 timesteps, along with the corresponding output probability of the guide policy and the actual sampled behavior policy. Except for employing the *Pick-Place* task's control policy during timesteps 20 to 30, the guide policy selects control policies of other tasks for the remaining timesteps, successfully accomplishing the task.

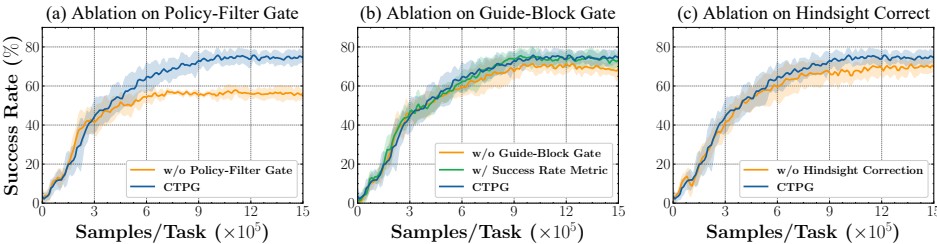

Figure 5: Three distinct ablation studies of MHSAC *w/* CTPG on MetaWorld-MT10.

## 5.3 Guidance Learned by Guide Policy

To answer **Q2**, we visualize the task *Pick-Place* with guidance on MetaWorld-MT10 to show the specific role of the guide policy. Since the guide policy is only used in trajectory generation during training, we visualize one of the sampled trajectories in Figure 4. We only present the initial 50 timesteps of this trajectory, in which the task has essentially been completed.

To better understand policy sharing between tasks, we first explain the relevant tasks. *Pick-Place*: Pick and place a puck to a goal. *Peg-Insert-Side*: Insert a peg sideways. *Push*: Push the puck to a goal. *Button-Press-Topdown*: Press a button from the top. *Drawer-Close*: Push and close a drawer. *Reach*: Reach a goal position. The visualizations of these tasks are provided in Figure 8 (Appendix D.1).

The initial and final 20 timesteps showcase that the guide policy learns useful guidance to successfully complete the source task. In the initial 20 timesteps, *Pick-Place* and *Peg-Insert-Side* employ a shared policy directing the robotic arm toward the target object. In the final 20 timesteps, the task is executed using *Button-Press-Topdown* to raise the gripper and then *Drawer-Close* to move forward. Interestingly, although the task in the final 20 timesteps intuitively aligns with *Reach*, the guide policy opts not to use it because the learned *Reach*'s control policy always opens the gripper, causing the puck to fall. In the middle 10 timesteps, the probability of *Pick-Place* is notably high due to the absence of alternative shared policies at this stage. Specifically, the most similar *Peg-Insert-Side* grabs the end of the peg for insertion, while *Pick-Place* requires gripping the puck centrally.

## 5.4 Ablation Studies

To answer **Q3**, we conduct detailed ablation studies to analyze the impact of each component in CTPG on performance enhancement. CTPG contains three key components: (1) the policy-filter gate, (2) the guide-block gate, and (3) the hindsight off-policy correction mechanism. We study the influence of each component using the MHSAC implicit knowledge sharing approach on MetaWorld-MT10.

(1) Figure 5(a) shows that the policy-filter gate plays a significant role within CTPG, leveraging the value function to constrain the direction of guided exploration. (2) Given the precise definition of binary-valued success signal in the MetaWorld benchmark, we compare the SAC temperature metric, described in Section 4.3, with another success rate metric to determine if tasks need guidance.

Specifically, we block guidance for tasks with success rates exceeding 80% during evaluation. Figure 5(b) illustrates that both metrics showcase competitive performance, exhibiting clear performance gains over no guide-block gate. However, since the success rate is a human-defined metric and difficult to define for some tasks (*e.g.*, HalfCheetah), the SAC temperature metric is more general across most environments. (3) We ablate the hindsight off-policy correction mechanism used in guide policy training in Figure 5(c), elucidating its improvement in training efficiency and stability. Further ablation studies of SM with CTPG on MetaWorld-MT50 are provided in Appendix E.2.

## 5.5 CTPG without Implicit Knowledge Sharing

To answer **Q4**, we train $N$ control and guide policies independently on $N$ tasks without implicit knowledge sharing. We perform experiments on HalfCheetah-MT8 and MetaWorld-MT10, employing the same environment configuration as in Section 5.2. As illustrated in Figure 6, CTPG significantly improves the performance of Single-Task SAC. Notably, the impact of CTPG on MetaWorld-MT10 is more pronounced. The potential rationale is that the tasks within HalfCheetah-MT8 exhibit minimal variance in difficulty, resulting in a narrow gap in

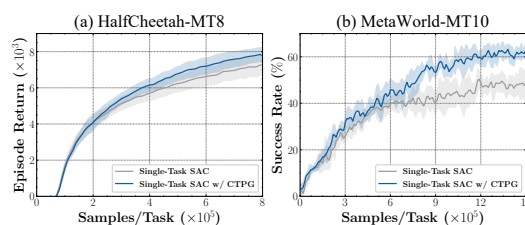

Figure 6: CTPG also improves performance in the absence of implicit knowledge sharing.

learning progress. Conversely, MetaWorld-MT10 presents a broader disparity in task difficulty, where CTPG facilitates the guidance from simpler to more challenging tasks, thus appearing more efficient.

## 5.6 Exploration of New Tasks with CTPG

To answer **Q5**, we split the original task set in half, pre-training expert policies on the one half $\mathbb{T}^e$. For the other half, we compare direct learning with CTPG, where the agent leverages guidance from both the control policies being learned and the expert policies. Specifically, we use MT-SAC on HalfCheetah-MT8, where $\mathbb{T}^e$ includes all tasks that enlarge the size of body parts. On the other hand, we use SM on MetaWorld-MT10, with $\mathbb{T}^e$ containing tasks indexed from 0 to 4. Figure 7 indicates that, compared to learning the

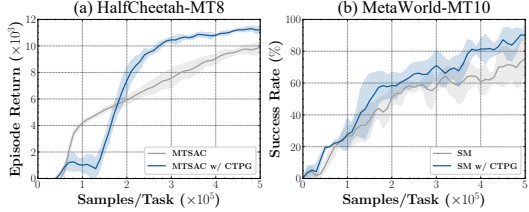

Figure 7: CTPG with expert policies can expedite the exploration of new tasks effectively.

new set of tasks from scratch, CTPG can reasonably utilize experts to explore new tasks rapidly. Notably, in environments where task similarity is high, such as HalfCheetah-MT8, CTPG can quickly transfer the expert's abilities to the control policy being learned with the guide policy.

## 6 Conclusion and Discussion

This paper proposes the Cross-Task Policy Guidance (CTPG) framework for the MTRL explicit policy sharing. CTPG contains a guide policy and two special gates to identify beneficial sharing policies from the set of all task control policies and choose the most proficient one to generate high-quality trajectories for the current task as policy guidance. In addition, CTPG is a generalized framework adaptable to diverse implicit knowledge sharing approaches. Empirical evidence showcases that these approaches combined with CTPG further improve sample efficiency and final performance.

**Limitations and Future Works.** One limitation of CTPG is its reliance on a predetermined guide step $K$, necessitating hyperparameter tuning for $K$ across different environments. Moreover, the fixed guide step setting lacks flexibility, as the duration of shared skills execution timesteps varies inconsistently among different tasks. Consequently, exploring methods to automate the selection of the guide step $K$ presents an intriguing avenue for future research. Additionally, irrespective of the human-defined win rate and the unique temperature parameter of SAC, investigating alternative metrics for the guide-block gate emerges as another important direction for future endeavors.

## 7 Acknowledgements

This work is supported in part by the National Science and Technology Major Project (2022ZD0116401), the Natural Science Foundation of China (Grant Nos. 62076238, 62222606, and 61902402), the Key Research and Development Program of Jiangsu Province (Grant No. BE2023016), and the China Computer Federation (CCF)-Tencent Open Fund.

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

# A  Pseudo Code

Without implicit knowledge sharing approaches, each task $i$ has its control policy $\pi_i$ and guide policy $\Pi_i^g$. With implicit knowledge sharing approaches, there is a unified control policy $\pi$ and guide policy $\Pi^g$ with the input of task representation $z_i$. In both cases, we uniformly represent tasks as subscripts and omit the network parameters for a clear presentation in pseudo-code. Specifically,

- The **actor network** with parameter $\phi$ for task $i$'s control policy is denoted as $\pi_i(a_t|s_t)$.
- The **critic network** with parameter $\theta$ for task $i$'s control policy is denoted as $Q_i(s_t, a_t)$.
- The **temperature parameter** for task $i$'s control policy is denoted as $\alpha_i$.
- The **guide actor network** with parameter $\phi^g$ for task $i$'s guide policy is denoted as $\Pi_i^g(j_t|s_t)$.
- The **guide critic network** with parameter $\theta^g$ for task $i$'s guide policy is denoted as $Q_i^g(s_t, j_t)$.
- The **guide temperature parameter** for task $i$'s guide policy is denoted as $\alpha_i^g$.
- The **comparable guide critic network** with parameter $\hat{\theta}^g$ for task $i$ is denoted as $\hat{Q}_i^g(s_t, j_t)$.

---

**Algorithm 1** Control Policy's Training Step

---

**Input**: minibatch $\mathcal{B}$ contains $\langle i, s_t, a_t, r_t, s_{t+1} \rangle$
**Initialization**: actor network $\pi$, critic network $Q$, SAC temperature $\alpha$

1: Optimize $\theta$ with SAC's critic loss in Equation 1

$$J_Q(\theta) = \mathbb{E}_{(i,s_t,a_t,r_t,s_{t+1})\sim\mathcal{B}} \left[ \frac{1}{2} \left( Q_i(s_t, a_t) - (r_t + \gamma V_i(s_{t+1})) \right)^2 \right]$$

2: Optimize $\phi$ with SAC's actor loss in Equation 3

$$J_\pi(\phi) = \mathbb{E}_{(i,s_t)\sim\mathcal{B}} \left[ \mathbb{E}_{a_t\sim\pi_i} \left[ \alpha_i \log \pi_i(a_t|s_t) - Q_i(s_t, a_t) \right] \right]$$

3: Optimize $\alpha$ with SAC's alpha loss in Equation 4

$$J(\alpha) = \mathbb{E}_{i\sim\mathcal{B}} \left[ \mathbb{E}_{a_t\sim\pi_i} \left[ -\alpha_i \log \pi_i(a_t|s_t) - \alpha_i \bar{\mathcal{H}} \right] \right]$$

---

**Algorithm 2** Guide Policy's Training Step

---

**Input**: actor network $\pi$, SAC temperature $\alpha$, minibatch $\mathcal{B}^g$ contains $\langle \{s_{t'}, a_{t'}, r_{t'}\}_{t'=t}^{t+K-1}, i, j_t, s_{t+K} \rangle$
**Initialization**: guide actor network $\Pi^g$, guide critic network $Q^g$, guide SAC temperature $\alpha^g$

1: Hindsight off-policy correct $j_t$ into $j_t'$ following Equation 9

$$j_t' = \arg\max_j \sum_{t'=t}^{t+K-1} \log \pi_j(a_{t'}|s_{t'})$$

2: Calculate the guide reward $r_t^g$ following Equation 6

$$r_t^g = \sum_{t'=t}^{t+K-1} \gamma^{t'-t} r_{t'}$$

3: Optimize $\theta^g$ with SAC's critic loss for guide policy

$$J_{Q^g}(\theta^g) = \mathbb{E}_{(i,s_t,j_t',r_t^g,s_{t+K})\sim\mathcal{B}^g} \left[ \frac{1}{2} \left( Q_i^g(s_t, j_t') - \left( r_t^g + \gamma^K V_i^g(s_{t+K}) \right) \right)^2 \right]$$

4: Optimize $\phi^g$ with SAC's actor loss for guide policy

$$J_{\Pi^g}(\phi^g) = \mathbb{E}_{(i,s_t)\sim\mathcal{B}^g} \left[ \mathbb{E}_{j_t\sim\Pi_i^g} \left[ \alpha_i^g \log \Pi_i^g(j_t|s_t) - Q_i^g(s_t, j_t) \right] \right]$$

5: Optimize $\alpha^g$ with SAC's alpha loss for guide policy

$$J(\alpha^g) = \mathbb{E}_{i\sim\mathcal{B}^g} \left[ \mathbb{E}_{j_t\sim\Pi_i^g} \left[ -\alpha_i^g \log \Pi_i^g(j_t|s_t) - \alpha_i^g \bar{\mathcal{H}}^g \right] \right]$$

---

## A.1 Pseudo Code of Policy Training

Algorithm 1 describes a training step for the control policy given a minibatch of data $\mathcal{B}$, which contains a batch of tasks $i$, states $s_t$, actions $a_t$, rewards $r_t$, and next states $s_{t+1}$.

In addition, Algorithm 2 describes a training step for the guide policy given a minibatch data $\mathcal{B}^g$. The data $\mathcal{B}^g$ not only contains a batch of tasks $i$, states $s_t$, actions $j_t$ taken by guide policy, and $K$ timesteps ahead states $s_{t+K}$ but also contains rewards $\{r_{t'}\}_{t'=t}^{t+K-1}$ accrued over $K$ timesteps for calculating the guide reward. Moreover, $\mathcal{B}^g$ includes states $\{s_{t'}\}_{t'=t}^{t+K-1}$ and actions $\{a_{t'}\}_{t'=t}^{t+K-1}$ taken by control policy over the $K$ timesteps for hindsight off-policy correction.

## A.2 Pseudo Code of Comprehensive CTPG

The full pseudo-code of CTPG is described in Algorithm 3.

---

**Algorithm 3** Cross-Task Policy Guidance

---

**Input**: guide step $K$
**Initialization**: actor network $\pi$, critic network $Q$, guide actor network $\Pi^g$, guide critic network $Q^g$, comparable guide critic network $\hat{Q}^g$, replay buffer $\mathcal{D} \leftarrow \emptyset$

1: **for** each epoch **do**
2:      Update the task subset $\mathbb{T}^g$ requiring guidance with

$$\mathbb{T}^g = \left\{ i \mid \log \alpha_i \leq \frac{1}{N} \sum_{j=1}^{N} \log \alpha_j \right\}$$

3:      // Inference Stage
4:      **for** each task $i$ **do**
5:          **for** each timestep $t = 0, 1, \ldots, T$ **do**
6:              **if** $t \% K = 0$ **then**
7:                  **if** $i \notin \mathbb{T}^g$ **then**                          ▷ Guide-Block Gate
8:                      Behavior policy $\pi' \leftarrow \pi_i$
9:                  **else**
10:                      Get comparable guide Q-values $\{\hat{Q}_i^g(s_t, j)\}_{j=1}^N$
11:                      Get control V-value $V_i(s_t)$ with Monte Carlo sampling control Q-values
12:                      Calculate policy-filter gate mask $m$ following          ▷ Policy-Filter Gate

$$m_j \leftarrow \begin{cases} 1, & \hat{Q}_i^g(s_t, j) \geq V_i(s_t) \\ 0, & \hat{Q}_i^g(s_t, j) < V_i(s_t) \end{cases}$$

13:                      Sample $\pi' \leftarrow \pi_{j_t}$ with $j_t \sim \text{Normalize}(\Pi_i^g(\cdot|s_t) \cdot m)$     ▷ Guide Policy
14:                  **end if**
15:              **end if**
16:              Execute action $a_t \sim \pi'(\cdot|s_t)$ and get reward $r_t$ and next state $s_{t+1}$
17:              Store the transition into $\mathcal{D}$
18:          **end for**
19:      **end for**
20:      // Training Stage
21:      **for** each training step $t = 0, 1, \ldots, T$ **do**
22:          Sample a minibatch $\mathcal{B}$ of all tasks from $\mathcal{D}$
23:          Optimize $\phi$ and $\theta$ with $\mathcal{B}$ as described in Algorithm 1
24:          **if** $t \% K = 0$ **then**
25:              Sample a minibatch $\mathcal{B}^g$ of task subset $\mathbb{T}^g$ from $\mathcal{D}$
26:              Optimize $\phi^g$ and $\theta^g$ with $\mathcal{B}^g$ as described in Algorithm 2
27:              Optimize $\hat{\theta}^g$ using Bellman Equation in Equation 13 with $\mathcal{B}^g$
28:          **end if**
29:      **end for**
30: **end for**

---

## B  Details on Comparable Guide Q-Value

We expand the value function of SAC to obtain:

$$
\begin{aligned}
V_i(s_t) &= \mathbb{E}_{a_t \sim \pi_i} \left[ Q_i(s_t, a_t) - \alpha_i \log \pi_i(a_t|s_t) \right] \\
&= \mathbb{E}_{a_t \sim \pi_i} \left[ Q_i(s_t, a_t) + \alpha_i \mathcal{H}(\pi_i(\cdot|s_t)) \right] \\
&= \mathbb{E}_{a_t \sim \pi_i} \left[ R_i(s_t, a_t) + \gamma \mathbb{E}_{s_{t+1} \sim P_i} \left[ V_i(s_{t+1}) \right] + \alpha_i \mathcal{H}(\pi_i(\cdot|s_t)) \right] \\
&= \mathbb{E}_{a_t \sim \pi_i} \left[ R_i(s_t, a_t) + \alpha_i \mathcal{H}(\pi_i(\cdot|s_t)) + \gamma \mathbb{E}_{s_{t+1} \sim P_i} \left[ V_i(s_{t+1}) \right] \right] \\
&= \cdots \\
&= \mathbb{E}_{a_{t'} \sim \pi_i, s_{t'+1} \sim P_i} \left[ \sum_{t'=t}^{\infty} \gamma^{t'-t} \left( R_i(s_{t'}, a_{t'}) + \alpha_i \mathcal{H}(\pi_i(\cdot|s_{t'})) \right) \right].
\end{aligned}
\tag{15}
$$

By repeatedly expanding $V_i$, we get the final form, which is actually the optimization objective under the maximum entropy reinforcement learning framework.

Then, we consider that the trajectory generation process solely using the current task's control policy $\pi_i$ can be regarded as equipped with a special guide policy $\Pi_i^{\tilde{g}}$ in SAC. The comparable guide Q-value in Equation 13 of this special guide policy $\Pi_i^{\tilde{g}}$ is:

$$
\begin{aligned}
\hat{Q}_i^{\tilde{g}}(s_t, i) &= \mathbb{E}_{a_{t'} \sim \pi_i, s_{t'+1} \sim P_i} \left[ \sum_{t'=t}^{t+K-1} \gamma^{t'-t} \left( R_i(s_{t'}, a_{t'}) + \alpha_i \mathcal{H}(\pi_i(\cdot|s_{t'})) \right) \right] + \gamma^K \mathbb{E}_{s_{t+K} \sim P_i} \left[ \hat{Q}_i^{\tilde{g}}(s_{t+K}, i) \right] \\
&= \mathbb{E}_{a_{t'} \sim \pi_i, s_{t'+1} \sim P_i} \left[ \sum_{t'=t}^{t+K-1} \gamma^{t'-t} \left( R_i(s_{t'}, a_{t'}) + \alpha_i \mathcal{H}(\pi_i(\cdot|s_{t'})) \right) + \gamma^K \hat{Q}_i^{\tilde{g}}(s_{t+K}, i) \right] \\
&= \cdots \\
&= \mathbb{E}_{a_{t'} \sim \pi_i, s_{t'+1} \sim P_i} \left[ \sum_{t'=t}^{\infty} \gamma^{t'-t} \left( R_i(s_{t'}, a_{t'}) + \alpha_i \mathcal{H}(\pi_i(\cdot|s_{t'})) \right) \right] \\
&= V_i(s_t).
\end{aligned}
\tag{16}
$$

The ellipsis part is the repeated expansion of $\hat{Q}_i^{\tilde{g}}$. Therefore, in SAC, the policy-filter gate can refine the action space of the guide policy by directly comparing $\hat{Q}_i^g(s_t, j_t)$ and $V_i(s_t)$.

## C  Details on Hindsight Off-Policy Correction for SAC

The hindsight off-policy correction mechanism is proposed to mitigate the non-stationarity challenge of the guide policy update process. It reassigns the action $j_t$ sampled by the past guide policy to a new one $j_t'$, whose control policy $\pi_{j_t'}$ is more likely to output the historical action sequence $\{a_{t'}\}_{t'=t}^{t+K-1}$. During the guide policy update process, we first get the reassigned action $j_t'$ following the maximum likelihood estimation in Equation 9. For SAC's critic update, the critic loss of the guide policy is,

$$
J_{Q^g}(\theta^g) = \mathbb{E}_{(i, s_t, j_t', r_t^g, s_{t+K}) \sim \mathcal{B}^g} \left[ \frac{1}{2} \left( Q_i^g(s_t, j_t') - \left( r_t^g + \gamma^K V_i^g(s_{t+K}) \right) \right)^2 \right],
\tag{17}
$$

where the hindsight correction explicitly modifies the update procedure by shifting the Q-value estimation from $Q_i^g(s_t, j_t)$ to $Q_i^g(s_t, j_t')$. For SAC's actor update, the actor loss of the guide policy is,

$$
J_{\Pi^g}(\phi^g) = \mathbb{E}_{(i, s_t) \sim \mathcal{B}^g} \left[ \mathbb{E}_{j_t \sim \Pi_i^g} \left[ \alpha_i^g \log \Pi_i^g(j_t|s_t) - Q_i^g(s_t, j_t) \right] \right],
\tag{18}
$$

which is not explicitly different from the original function in SAC. However, it indirectly affects actor learning due to SAC's unique actor update method [9]. Specifically, SAC's actor loss is derived from its optimization objective, and the policy is updated according to,

$$
\pi_{\text{new}} = \arg \min_{\pi' \in \Pi} D_{KL} \left( \pi'(\cdot|s_t) \left\| \frac{\exp \left( \frac{1}{\alpha} Q^{\pi_{\text{old}}}(s_t, \cdot) \right)}{Z^{\pi_{\text{old}}}(s_t)} \right. \right),
\tag{19}
$$

where the partition function $Z^{\pi_{\text{old}}}(s_t)$ normalizes the distribution. In essence, SAC's policy is updated by fitting the distribution of the SoftMax function of Q-value with temperature $\alpha$. Therefore, modifying the update of $Q_i^g(s_t, j_t)$ to $Q_i^g(s_t, j_t')$ using hindsight correction leads to a different guide policy actor optimization objective, thus affecting the training of the guide actor network.

# D Environment Details

## D.1 MetaWorld Manipulation Benchmark

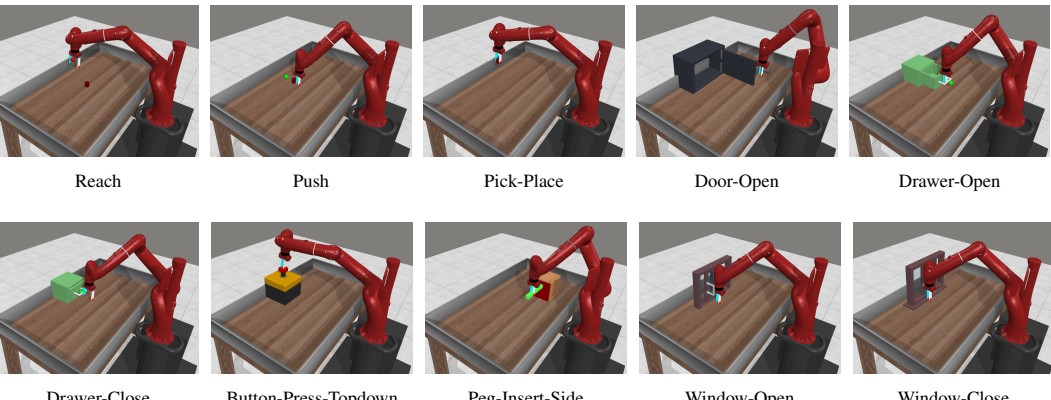

Figure 8: Visualizations of robotic manipulation tasks on MetaWorld-MT10.

The MetaWorld manipulation benchmark [31] consists of 50 robotics manipulation tasks employing a sawyer arm to acquire diverse manipulation skills. *MetaWorld-MT50* encompasses all 50 manipulation tasks, and *MetaWorld-MT10* encompasses 10 tasks (a subset of MT50), as illustrated in Figure 8. In MetaWorld, the source tasks are configured with fixed goals, limiting the policy's ability to generalize to tasks of the same type with varying goals. Following a similar setup as [27], we extend all the tasks to a random-goal setting, where both items and goals reset randomly at each episode's onset. In addition, unlike [27, 30], we conduct our experiments on the MetaWorld-V2 benchmark [2].

In addition, we extend the episode length to 200 timesteps, different from the previous setting of 150. We evaluate the rule-based policies provided by the MetaWorld benchmark across 100 sample episodes. Under the initial episode length setting of 150 timesteps, 42 tasks achieve a success rate exceeding 90%. However, among the remaining 8 tasks, the task *Disassemble* exhibits a mere 50% success rate. In contrast, upon adjusting the episode length setting to 200 timesteps, success rates for all 50 tasks exceed 90%, with the task *Disassemble* achieving an improved success rate of 91%.

## D.2 HalfCheetah Locomotion Benchmark

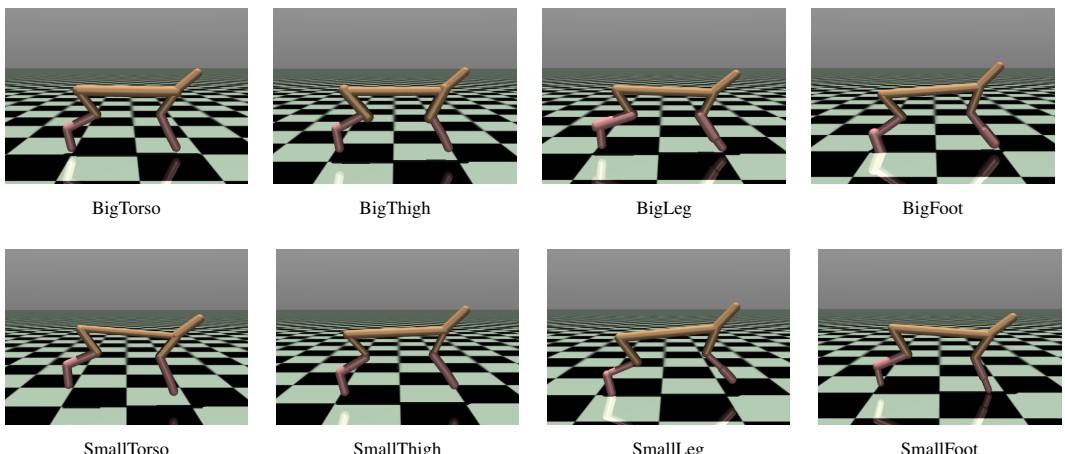

Figure 9: Visualizations of robotic locomotion tasks on HalfCheetah-MT8.

---

[2]https://github.com/Farama-Foundation/Metaworld/tree/v2.0.0

The HalfCheetah locomotion benchmark originates from the gym-extensions framework [11] [3], which focuses on continuous control multi-task reinforcement learning tasks, including several task groups of the standard gym environments [2]. We run our experiments with the HalfCheetah agent, which is a 2-dimensional robot consisting of 9 links and 8 joints connecting them (including two paws). Each episode contains 1000 timesteps.

**HalfCheetah-MT5** consists of 5 locomotion tasks. The HalfCheetah agent is tasked with running in environments with various scales of simulated earth-like gravity, ranging from one-half to one-and-a-half of the normal gravity level. The detailed gravity value of each task is illustrated in Table 2.

Table 2: The gravity setup of HalfCheetah-MT5.

| Task Name | Gravity ($m \cdot s^{-2}$) |
|---|---|
| GravityHalf | 4.91 |
| GravityThreeQuarters | 7.36 |
| GravityNormal | 9.81 |
| GravityOneAndQuarter | 12.26 |
| GravityOneAndHalf | 14.72 |

**HalfCheetah-MT8** consists of 8 locomotion tasks, as depicted in Figure 9. The HalfCheetah agent must run with variations in the morphology of a specific body part, such as the torso, thigh, leg, and foot. The *Big* body part involves scaling the mass and width of the limb by 1.25, and the *Small* body part involves scaling by 0.75.

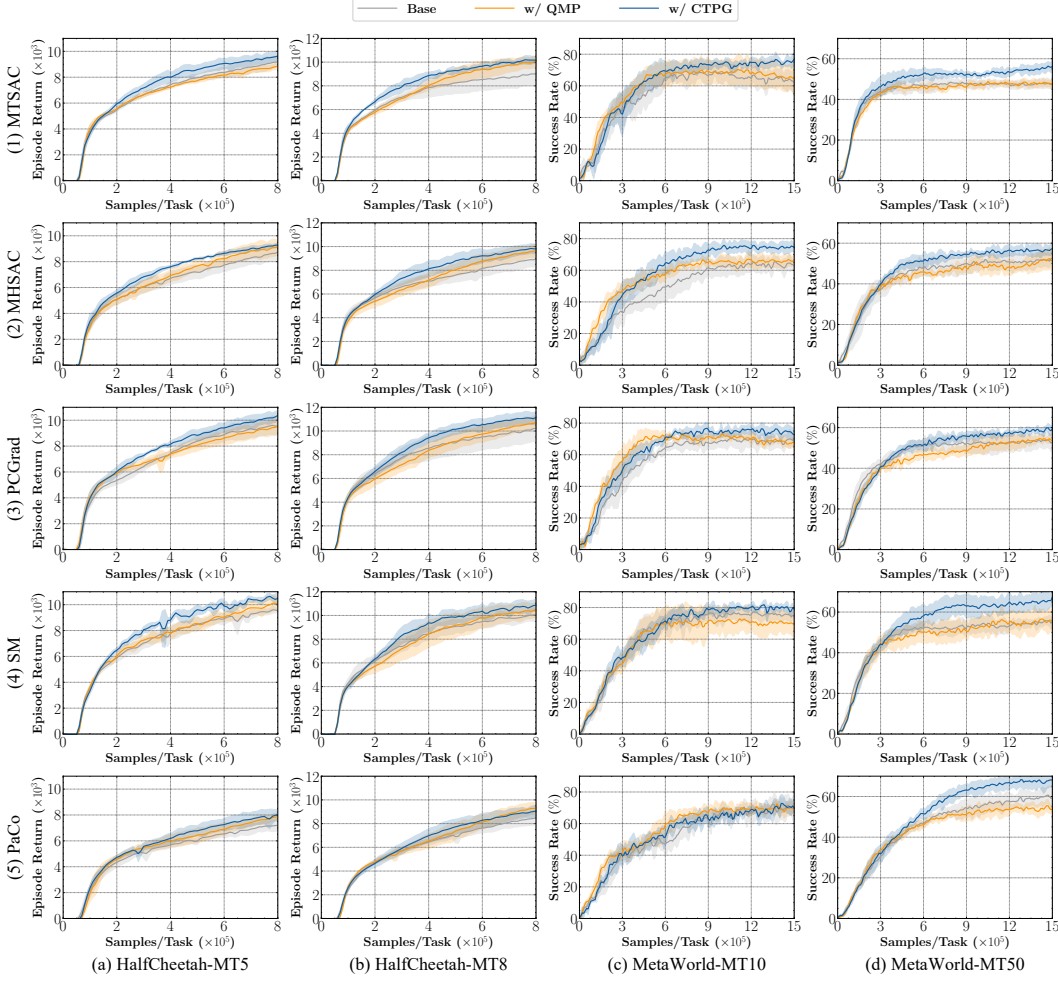

Figure 10: Training curves of experiment with implicit knowledge sharing approaches. Beyond the ultimate performance improvement, CTPG also enhances the sample efficiency.

---

[3]https://github.com/Breakend/gym-extensions

# E Additional Experimental Results

## E.1 Training Curves of Experiment with Implicit Knowledge Sharing Approaches

This section complements the result in Section 5.2, which only presents the final performance of the experiment. Here, we show the training curves for different combinations of explicit policy sharing methods and implicit knowledge sharing approaches across four environments in Figure 10. Each row of the figure represents a distinct implicit knowledge sharing approach, while each column represents a different environment. Within each subfigure, the three curves represent the base one without any explicit policy sharing method and two variations using different explicit policy sharing methods. We evaluate the training policy every 10K samples per task with 32 episodes and report the mean episode return or success rate, along with standard deviation, across 5 different seeds. The result shows that beyond the ultimate performance improvement, CTPG also enhances the sample efficiency.

## E.2 Additional Results of Ablation Studies

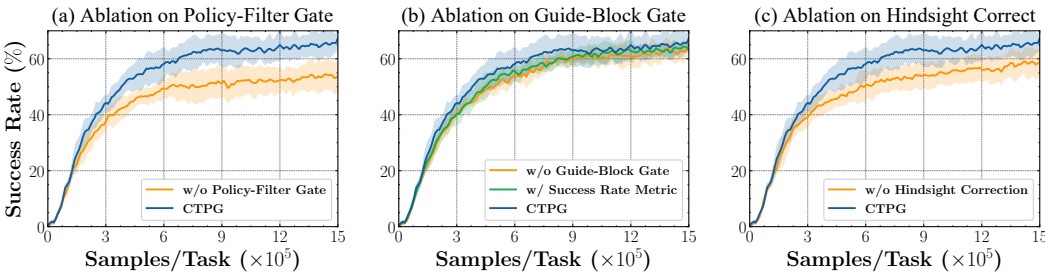

Figure 11: Three distinct ablation studies of SM *w/* CTPG on MetaWorld-MT50.

In addition to the ablation experiment of MHSAC with CTPG on MetaWorld-MT10 in Section 5.4, we also conduct ablation studies using SM implicit knowledge sharing approach on MetaWorld-MT50. Figure 11 shows similar results to those in Section 5.4. The policy-filter gate is crucial within CTPG. The guide-block gates with two metrics demonstrate competitive performance and show improvement compared to the absence of the guide-block gate. Additionally, the hindsight off-policy correction mechanism significantly enhances training efficiency and stability.

## E.3 Ablation Study on Guide Step

Within CTPG, the guide policy $\Pi_i^g$ selects a policy $\pi_{j_t}$ from the candidate set of all control policies $\{\pi_j\}_{j=1}^N$ every fixed $K$ timesteps. The selected policy $\pi_{j_t}$ is then used as the behavior policy to interact with the environment and collect data for the next $K$ timesteps.

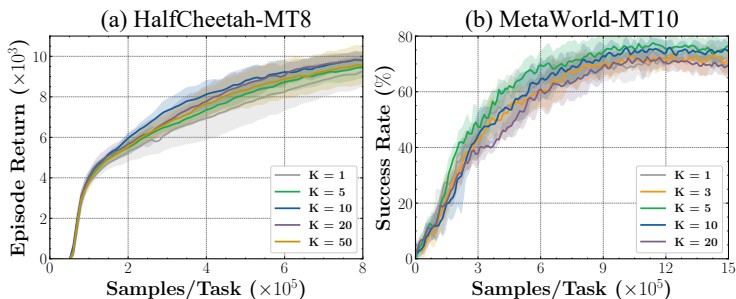

Figure 12: MHSAC *w/* CTPG with different guide steps $K$.

The guide step $K$ is a predefined hyper-parameter. We conduct ablation experiments on the guide step $K$ in the two setups: HalfCheetah-MT8 and MetaWorld-MT10. In HalfCheetah-MT8, an entire episode contains 1000 timesteps, so we set $K \in \{1, 5, 10, 20, 50\}$. In MetaWorld-MT10, an entire episode contains 200 timesteps, so we set $K \in \{1, 3, 5, 10, 20\}$. The result, shown in Figure 12, indicates that both short and long guide steps lead to decreased performance and increased variance.

Overall, CTPG performs well in both environments when $K = 10$. Furthermore, automating the selection of the guide step $K$ is part of our future work.

## E.4 Ablation Study on Monte Carlo Sampling

In Section 4.2, we design a policy-filter gate to refine the action space of the guide policy by adaptively filtering out control policies that are not beneficial for guidance. Specifically, the restriction of the action space is implemented by a mask, which is generated by comparing the control policy's V-value $V_i(s_t)$ and the guide policy's Q-value $Q_i^g(s_t, j_t)$.

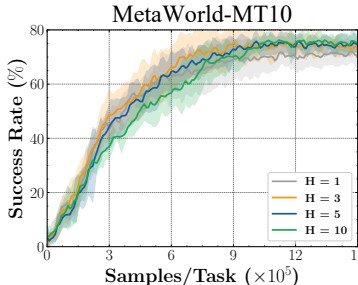

Figure 13: MHSAC *w/* CTPG with various Monte Carlo sampling times $H$ to estimate V-value.

In SAC, the V-value is the expectation of the Q-value with the entropy of policy. It is formulated as:

$$V_i(s_t) = \mathbb{E}_{a_t \sim \pi_i} \left[ Q_i(s_t, a_t) - \alpha_i \log \pi_i(a_t|s_t) \right] \tag{20}$$

To avoid introducing an additional network to estimate the V-value $V_i(s_t)$, we estimate it via Monte Carlo sampling of the Q-values $Q_i(s_t, a_t)$ with $a_t \sim \pi_i(a_t|s_t)$ [15]. We conduct ablation experiments varying the Monte Carlo sampling times $H$, as depicted in Figure 13. It is observed that the performance remains largely unaffected by the number of samples, except in cases where $H$ equals 1, which leads to performance degradation and heightened variance due to excessive randomness.

## E.5 Additional Comparison with BPT

CTPG learns a flexible mixture policy combining different control policies. To verify the advantages of the mixture strategy, we compare CTPG against another baseline that incorporates a guide strategy from [26], which solely learns from another control policy with the highest performance on the task. Since [26] focuses on continual RL, which requires a predefined task sequence and does not align with the MTRL experimental setting. Therefore, we adapt the core idea, implementing a version that selects the best-performing policy for the current task to guide exploration and generate trajectory data. Specifically, after each $H$ rounds of data collection, we evaluate all policies across all tasks, selecting the top-performing policy for each task to guide data collection in the subsequent rounds. We refer to this baseline as BPT (Best Performance Transfer) and set $H$ to 50 episodes in our implementation. We use MTSAC in HalfCheetah-MT8 and MHSAC in MetaWorld-MT10.

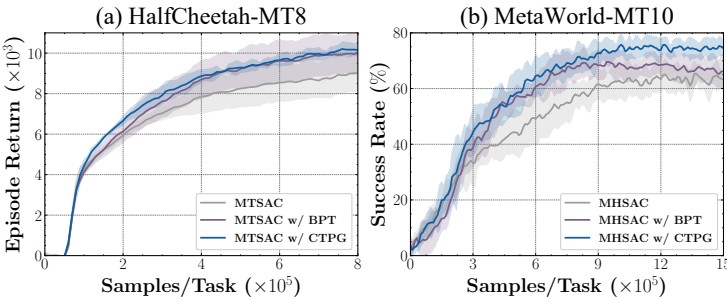

Figure 14: The result of additional comparison with BPT.

The result, shown in Figure 14, demonstrates that CTPG outperforms BPT. CTPG's guide policy learns a more flexible strategy: it not only can learn to share a single policy within a complete

trajectory (like BPT), but also can develop a mixture policy by combining different control policies, making it more transferable between tasks. Notably, BPT performs better in HalfCheetah-MT8 than in MetaWorld-MT10 because the tasks in HalfCheetah-MT8 share more significant similarities, whereas the policy transfer and guidance in MetaWorld-MT10 require combination strategies.

### E.6 Adaptability of Other RL Algorithms to CTPG

CTPG is a general MTRL framework that can be adapted with other RL algorithms, even allowing the control and guide policy to use different RL algorithms. Since SAC is a widely used algorithm in continuous control and serves as the base algorithm for all baselines, we also choose SAC as the base algorithm in this work. In addition, we explore the adaptation of TD3 with the CTPG and also employ different RL algorithms for the control and guide policies. Specifically, the control policy uses TD3, and the guide policy uses DQN. We evaluate MTTD3 (the original TD3 using one-hot encoding for task representation) on HalfCheetah-MT8 and MHTD3 (utilizing a multi-head network for different tasks) on MetaWorld-MT10. The result shown in Figure 15 demonstrates that CTPG can enhance performance and sample efficiency when combined with other backbone RL algorithms.

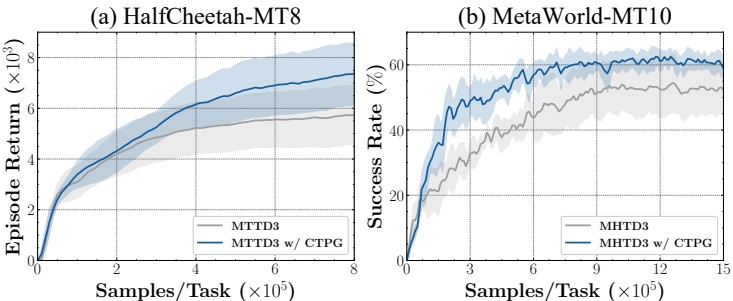

Figure 15: The result of CTPG based on TD3 RL algorithm.

## F Implementation Details

We implement all experiments using the MTRL codebase [21] [4] and access the HalfCheetah locomotion environment to this framework. One significant change we make involves removing the reward normalizer wrapper, which leads to worse results.

### F.1 Details on Computational Resources

We use AMD EPYC 7742 64-Core Processor with NVIDIA Geforce RTX 3090 GPU for training. Each method is trained 5 times with different seeds. Using MetaWorld-MT10 as an example, the training time required for a full training run varies from 16 hours (MTSAC) to 52 hours (PcGrad) for the baseline implicit knowledge sharing approaches. For each approach with CTPG, the training times range from 20 hours (MTSAC *w/* CTPG) to 56 hours (PcGrad *w/* CTPG). The guide policy, using a simple multi-head network architecture, acts every $K$ step ($K = 10$ in our implementation), so its training frequency is set as $1/K$. Consequently, CTPG does not significantly increase the training time. For a detailed hyper-parameter of the guide policy, please refer to Appendix F.3.

### F.2 Additional MTRL Training Setups

For a fair comparison, we use the following common training setups on all methods:

**Disentangled SAC temperature parameters.** This setup is standard established in MTRL. Herein, distinct tasks utilize varying SAC temperature parameters, as described in Equations 2 and 3, which are also optimized by independent losses in Equation 4. Therefore, this setup facilitates the individual adaptation of exploration and exploitation balances for each task, effectively accommodating the diverse learning dynamics across different tasks during training.

---

[4]https://github.com/facebookresearch/mtrl

**Loss maskout of extreme tasks.** This setup, utilized by [21, 22], involves the selective masking of the potentially destabilized loss $J_i$ of task $i$ from the total loss $J$, aiming to mitigate its adverse effects on other tasks. Specifically, when the task loss $J_i$ surpasses a predefined threshold $\epsilon$ (set as $3e3$, the same as [22]), that task $i$ is excluded from the total training loss.

**Multi-task loss rescaling.** In recognition of the inherent discrepancy in convergence rates between tasks, where easy tasks usually converge faster, [27] propose an optimization objective weight of task $i$, formulated as:

$$w_i = \frac{\exp(-\alpha_i)}{\sum_{j=1}^{N} \exp(-\alpha_j)}, \tag{21}$$

where $\alpha_i$ is the SAC temperature parameter of task $i$, and $N$ is the total number of tasks. Consequently, the total loss is adjusted to $J = \mathbb{E}_i[w_i \cdot J_i]$, ensuring a balanced training process across different tasks.

### F.3 Hyper-Parameters of All Method

This section provides the hyper-parameters of each method in our experiment. General hyper-parameters shared by all methods are illustrated in Table 3. Table 4 to Table 8 show the additional hyper-parameters specific to each implicit knowledge sharing approach. In addition, the guide policy can also be trained using implicit knowledge sharing methods. In our implementation, we use the simple multi-head network structure for the guide policy in CTPG, and the relevant additional hyper-parameters are displayed in Table 10.

Table 3: General hyper-parameters of all methods.

| Hyper-parameter | Value |
|---|---|
| network architecture | feedforward network |
| batch size | 128 × number of tasks |
| non-linearity | ReLU |
| policy initialization | standard Gaussian |
| # of samples / # of train steps per iteration | 1 env step / 1 training step |
| policy learning rate | 1e-4 |
| Q function learning rate | 1e-4 |
| alpha learning rate | 1e-4 |
| optimizer | Adam |
| discount | .99 |
| episode length | 1000 (HalfCheetah) / 200 (MetaWorld) |
| exploration steps | 2000 |
| reward scale | 0.1 |
| replay buffer size | 1e6 (MT5 / MT8 / MT10) / 1e7 (MT50) |

Table 4: Additional hyper-parameters of MTSAC.

| Hyper-parameter | Value |
|---|---|
| network hidden layer | 2 (HalfCheetah) / 5 (MetaWorld) |
| network hidden size | 400 |

Table 5: Additional hyper-parameters of MHSAC.

| Hyper-parameter | Value |
|---|---|
| network architecture | multi-head (1 head / task) |
| network hidden layer | 2 (HalfCheetah) / 5 (MetaWorld) |
| network hidden size | 400 |

Table 6: Additional hyper-parameters of PCGrad.

| Hyper-parameter | Value |
| --- | --- |
| optimizer | Adam *w/* PCGrad |
| network hidden layer | 2 (HalfCheetah) / 5 (MetaWorld) |
| network hidden size | 400 |

Table 7: Additional hyper-parameters of SM.

| Hyper-parameter | Value |
| --- | --- |
| state representation | [400] (HalfCheetah) / [400, 400] (MetaWorld) |
| task representation | [400] |
| number of layers | 4 |
| number of modules/layer | 4 |
| module size | 64 (HalfCheetah) / 128 (MetaWorld) |
| routing size | 128 |

Table 8: Additional hyper-parameters of Paco.

| Hyper-parameter | Value |
| --- | --- |
| parameter set number | 3 |
| network hidden layer | 2 (HalfCheetah) / 3 (MetaWorld) |
| network hidden size | 400 |

Table 9: Additional hyper-parameters of QMP.

| Hyper-parameter | Value |
| --- | --- |
| temporally extended length | 10 |

Table 10: Additional hyper-parameters of guide policy in CTPG.

| Hyper-parameter | Value |
| --- | --- |
| network architecture | multi-head (1 head / task) |
| network hidden layer | 2 (HalfCheetah) / 5 (MetaWorld) |
| network hidden size | 400 |
| guide step | 10 |
| # of samples / # of train steps per iteration | 10 env step / 1 training step |
| Monte Carlo sampling time | 5 |
| guide policy learning rate | 1e-4 |
| guide Q function learning rate | 1e-4 |
| guide alpha learning rate | 1e-4 |

# G   Broader Impacts

This paper presents work that aims to advance the field of Multi-Task Reinforcement Learning. There are many potential societal consequences of our work, none of which we feel must be specifically highlighted here.

