# OpenReview forum: "Efficient Multi-task Reinforcement Learning with Cross-Task Policy Guidance"
_NeurIPS.cc/2024/Conference — NeurIPS 2024 poster_

### Official Review · Reviewer_k7nL · 2024-07-10

**Soundness:** 2
**Presentation:** 3
**Contribution:** 2
**Rating:** 6
**Confidence:** 4

**Summary:**

The paper presents Cross-Task Policy Guidance (CTPG), a novel framework designed to improve multi-task reinforcement learning (MTRL) by leveraging cross-task policy similarities. CTPG trains a guide policy for each task to select the most suitable behavior policy from a pool of all tasks' control policies. This method aims to generate better training trajectories and enhance learning efficiency. The authors propose two gating mechanisms: a policy-filter gate to filter out non-beneficial control policies and a guide-block gate to block unnecessary guidance for mastered tasks. Empirical evaluations on manipulation and locomotion benchmarks demonstrate that integrating CTPG with existing parameter sharing approaches significantly enhances performance.

**Strengths:**

### S1. Innovative Approach

The paper introduces a novel method for leveraging cross-task similarities in MTRL. By training a guide policy to select behavior policies from a pool of tasks, the approach provides a direct and efficient way to exploit shared skills, which is an underexplored area in MTRL.

### S2. Empirical Validation

The proposed framework is validated through extensive experiments on various benchmarks, including MetaWorld and HalfCheetah. The results demonstrate significant improvements in learning efficiency and performance when CTPG is integrated with existing MTRL approaches.

### S3. Clear Presentation

The paper is well-structured and clearly explains the CTPG framework, including detailed descriptions of the guide policy and gating mechanisms. Figures and tables effectively illustrate the benefits of the proposed method, and the provided pseudocode enhances reproducibility.

**Weaknesses:**

### W1. Scalability Concerns

The scalability of CTPG to more complex and large-scale environments is not thoroughly discussed. While the method shows promising results in the tested benchmarks, a broader analysis of its scalability and practical utility in more complex scenarios may improve the significance of this work.

### W2. Computational Complexity

The computational complexity of training and deploying CTPG, particularly the guide policy and the gating mechanisms, is not explicitly addressed. Understanding the computational requirements and potential limitations in terms of resources and execution time would provide a more comprehensive evaluation of its applicability.

**Questions:**

### Q1. Scalability to Complex Environments

How does the CTPG framework scale to more complex, large-scale environments? Are there any specific challenges or limitations that need to be addressed for practical deployment in such scenarios?

### Q2. Computational Requirements

What are the computational requirements for training and deploying CTPG? How does the method perform in terms of execution time and resource consumption compared to existing MTRL methods?

**Limitations:**

The authors acknowledge the limitations related to the predetermined guide step K and the reliance on specific benchmarks. However, a more detailed discussion on potential negative societal impacts and strategies to mitigate them would be beneficial.

---

> ### Author Rebuttal · Authors · 2024-08-06
>
> Thank you very much for your constructive feedback and acknowledgment of our efforts. Following are our responses to all your concerns.
>
> > Scalability to complex environments.
>
> We have tested CTPG on MetaWorld and HalfCheetah, the commonly used and authoritative benchmarks in the MTRL community, covering two main fields of manipulation and locomotion. In both two environments with diverse task settings, CTPG has shown outstanding performance, which demonstrated that CTPG can effectively learn policy sharing between tasks, thus improving the sample efficiency of MTRL. In addition, CTPG is a general MTRL framework that can be adapted to other MTRL environments and other RL algorithms. We conducted experiments using another base RL algorithm TD3 in **Fig.3 of the global Rebuttal Comment PDF**, and the results demonstrate that CTPG is equally applicable to other algorithms to enhance sample efficiency. CTPG can also be migrated to other MTRL benchmarks without additional configuration and deployment. If you have a recommendation for another benchmark, we'd be happy to try it on there too :)
>
> > Computational requirements and computational complexity.
>
> We provided detailed information about the computational resources used in our experiments in Appendix E.1. For computational complexity, we show the training curves with the x-axis being wall-clock time on the MetaWorld-MT10 environment in **Fig.4 of the global Rebuttal Comment PDF**. The results indicate that although CTPG takes longer to train, it still outperforms baselines for the same training time. Furthermore, during the evaluation and deployment phase, CTPG's guide policy will no longer be used, so no additional execution time and storage space is required.
>
> > Discussion on potential negative societal impacts.
>
> We provided Broader Impacts in Appendix F. CTPG is a general MTRL framework and does not introduce additional social impacts that need to be discussed. If you feel there are any necessary social implicates that need to be discussed, welcome to point them out, and we will add discussions about them in the revised manuscript.
>
> **Thank you once again for your valuable feedback. We hope our response has satisfactorily addressed your concerns. If you find that we have addressed your concerns, we kindly hope you reconsider your rating. If you need further elaboration or additional points to include in the response, we welcome further discussion to ensure everything is clear and satisfactory.**

---

> > ### Comment · Reviewer_k7nL · 2024-08-10
> > **Response to authors rebuttal**
> >
> > I appreciate the authors for the detailed response. Most of my concerns are addressed therefore I'm raising my assessment.

---

> > > ### Author Response · Authors · 2024-08-10
> > >
> > > Thank you very much for reviewing our response. We are pleased that our response has effectively addressed your concerns. Once again, we sincerely appreciate your thorough review and insightful feedback.
> > >
> > > Best regards,
> > >
> > > All authors

---

### Official Review · Reviewer_J1dv · 2024-07-11

**Soundness:** 3
**Presentation:** 3
**Contribution:** 2
**Rating:** 5
**Confidence:** 4

**Summary:**

This paper tackles the problem of multi-task reinforcement learning. Previous works have primarily focused on MTRL through specialized network structures or methods to resolve conflicting gradients. This paper considers the problem from an orthogonal perspective, by actively sharing control policies for each task adaptively, in the hope that sometimes the policy of one task can guide the exploration of other tasks. Specifically, similar to in hierarchical RL, a high-level policy is used to decide which “task policy” to call at each particular state, where “bad” policies are masked out through a learned value function. Evaluated on Metaworld and HalfCheetah MTRL benchmark, the method shows improved performance compared to previous works.

**Strengths:**

- The idea of exploring with cross-task policies to facilitate multi-task reinforcement learning is interesting and novel.
- The method is thoroughly evaluated on a set of test domains, and show superior performance compared to previous methods
- This paper is well written, and the method is easy to follow.

**Weaknesses:**

- The proposed method contains many floating pieces that seem to add burden both to implementation and hyperparameter tuning. Some of the pieces seem rather like hack than a principled solution. For example:
  - For the hindsight off-policy correction, the action of the guide policy is relabelled to the one that is most likely to generate the sequence of actions. However, it should be likely that none of the control policies can generate the old sequence of actions with high probability, in which case relabeling will not help at all.
  - The action space masking is essentially masking out control policies with low Q values. Shouldn’t this already be reflected in the policy through policy gradient updates? Why do we need additional masking?
  - Using the temperature coefficient of SAC as an indication of the policy performance is very empirical and may not always be true. This also makes the proposed method specific to adaptive temperature SAC.
- It seems that the hindsight off-policy correction would require much more forward passes than a regular SAC. Can the author show the performance of the proposed method with x axis being wall-clock time?
- For the proposed method to be effective, we need the policy of some other tasks to have better performance than the policy that is trained for the current task. One scenario I can think of is when the multiple tasks secretly form a curriculum (e.g. one task is the prerequisite for another task), which might be the case in Metaworld and halfcheetah. But I’m not sure how realistic such an assumption is in the real world.

Minors:
- Line 207: dose → does
- “someone who can ride a bicycle can quickly learn to ride a motorcycle by referring to related skills” → I don’t think that’s actually the case…

**Questions:**

See weaknesses

**Limitations:**

The limitations are adequately addressed.

---

> ### Author Rebuttal · Authors · 2024-08-06
>
> Thank you very much for your constructive feedback and acknowledgment of our efforts.
>
> > Floating pieces
>
> 1. Hindsight off-policy correction: On the one hand, it is deeply integral to our CTPG framework, instead of just a floating piece. It addresses the unavoidable non-stationarity issue in off-policy training. We demonstrate this in Fig 5c and 11c, where it is helpful and its effect is more pronounced with more tasks, reinforcing its necessity within CTPG. On the other hand, it seems that there is a misunderstanding on the reason for proposing this module. The issue you mentioned does indeed exist and is still a matter of interest for the community. In such case, the data is invalid for training because none of the policies can regenerate this old sequence, so the hindsight off-policy correction won't work either. However, our module is not proposed to solve this issue and has its own motivation. It should not be considered a floating piece merely because it does not address a problem outside its intended scope.
>
> 2. Action space masking in the policy-filter gate: The integration of this module with CTPG is also essential, which avoids negative policy guidance from unhelpful control policies. Although the guide policy can also be learned by gradient updates only, the policy-filter gate serves to expedite this process by narrowing down the candidate policies, especially when the number of tasks is particularly large, thus further reducing the exploration space and improving sample efficiency. As shown in Fig 5a and 11a, training efficiency will be significantly reduced without this gate. Here we want to emphasize that although the functionality of this gate can be achieved by policy gradient update, our module can significantly improve training efficiency.
> 3. Temperature coefficient of SAC in the guide-block gate: We aim to use it to filter tasks that have been mastered or converged on. The choice of SAC and its adaptive temperature mechanism was made due to its robust performance in various RL tasks. We want to emphasize that CTPG is a general MTRL framework that can be adapted with other RL algorithms. We explore the adaptation of TD3 in **Fig.3 of the global Comment PDF**. Using the temperature of SAC as an indicator is indeed an empirical choice. We admit it may not always be true, thus we provide several alternatives not specific to SAC. For example, we try an intuitive metric success rate in Fig 5b and 11b, which shows fair performance with SAC temperature. Besides, policy entropy (in algorithms like PPO) or cumulative rewards (when there is no significant change) can also perform as indicators. The guide-block gate serves as an essential module with various implementations, and we provide several alternatives and welcome further study based on ours.
>
> The three modules collectively contribute to share effective policies between tasks, thus improving the sample efficiency, which is the key goal of MTRL. In addition, these three modules do not involve any hyperparameter settings, so there is no additional burden for tuning. Hope our response can change your initial perception that they were just floating pieces.
>
> > Forward time of hindsight off-policy correction and performance with wall-clock time
>
> Compared to the gradient update process, hindsight off-policy correction requires only forward computation, making the time consumed negligible. In addition, the calculation of each control policy's probability for generating the historical action sequence can be parallelized. This is achieved by extending the input of the actor from [batch\_size] to [batch\_size, task\_num], where the second dimension distinguishes the different task representations.
>
> **Fig.4 of the global Comment PDF** shows the training curves with wall-clock time on MetaWorld-MT10. Although CTPG takes longer to train, it still outperforms baselines for the same training time.
>
> > Scenarios that CTPG works
>
> CTPG does not require that a task have to be a sub-strategy of another task. If tasks can naturally form a curriculum as you mentioned, CTPG definitely can learn an effective policy. Even if other tasks' control policies do not perform as well on the current task, CTPG's guide policy can still facilitate an effective mixed policy. In specific, the agent can choose different control policies at different stages within a single episode, as shown in Fig 4 in the paper. In other words, the policy guidance uses a learnable mixture policy consisting of all control policies. Furthermore, referring to Reviewer xTD7's suggestion, we set up a new baseline called BPT (Best Performance Transfer), which selects the best-performing policy in the current task among all task policies to guide the exploration. The results shown in **Fig.1 of the global Comment PDF** demonstrate that CTPG also outperforms BPT. This is because CTPG's guide policy learns a more flexible strategy, rather than simply relying on a single best-performing policy for guidance.
>
> Here we extend the earlier example involving bicycle and motorcycle for better understanding. Suppose an agent is learning four tasks: unicycle(U), bicycle(B), motorcycle(M), and automobile(A). They do not form a curriculum and need to be learned simultaneously. (B) and (M) share the skill balancing on two wheels. (U) and (B) share the skill pedal-driven, while (M) and (A) share the skill ignition to move. With CTPG, if the agent has learned the pedaling skill in (B), it can transfer this skill to (U), even if it hasn’t yet mastered balancing. Similarly, if the agent has learned balance in (B) and ignition in (A), it can quickly master (M) by combining these two skills.
>
> > Minor errors
>
> Thank you for pointing out the typo "dose", we will correct it in the revised manuscript.
>
> **Thank you once again for your valuable feedback. If you need further elaboration or additional points to include in the response, we welcome further discussion to ensure everything is clear and satisfactory.**

---

> > ### Comment · Reviewer_J1dv · 2024-08-09
> >
> > I thank the authors for the clarifications. I will maintain my original score.

---

> > > ### Author Response · Authors · 2024-08-09
> > >
> > > We greatly appreciate your time. If there are any remaining issues or concerns that have not been addressed, please let us know. We would be more than happy to engage in further discussion to resolve them. If all concerns have been addressed, we kindly hope you can reconsider the rating assigned to our submission.
> > >
> > > Thank you once again for your valuable input.
> > >
> > > Best regards,
> > >
> > > All authors

---

### Official Review · Reviewer_5DoG · 2024-07-12

**Soundness:** 2
**Presentation:** 2
**Contribution:** 2
**Rating:** 5
**Confidence:** 5

**Summary:**

This paper addresses the problem of multi-task reinforcement learning (MTRL). To this end, the paper proposes a method that selectively shares behaviors from the policies learning to solve other tasks. The experiments conducted in a locomotion domain (multi-task Half-Cheetah) and a robot arm manipulation domain (Meta-World) verify that the proposed method can improve the performance of the learned policies. Ablation studies justify the effectiveness of many proposed components, including the policy-filter gate, the guide-block gate, and the hindsight correct. Yet, I am not convinced that the proposed method could improve the sample efficiency, which is the key goal of MTRL. Also, it seems that the proposed method is incremental and lacks novelty. Therefore, I am leaning toward rejecting this work in its current form.

**Strengths:**

**Motivation and intuition**
- The motivation for sharing behaviors for MTRL is convincing.

**Experimental results**
- The main experimental results show that the proposed method achieves better converged performance compared to the baselines in MT Half-Cheetah and Meta-World.

**Ablation study**
- The ablation studies justify the effectiveness of the policy filter gate, guide block gate, and correct hindsight.

**Reproducibility**
- The code is provided, which helps understand the details of the proposed method.

**Weaknesses:**

**Main results (Table 1)**
- Table 1 and Section 5.2 present quantitative results of converged performance. However, as far as I am concerned, the comparisons of MTRL should focus on sample efficiency, i.e., how fast each method can learn, instead of converged performance. Therefore, I am not convinced by this evaluation.

**Comparison to k-step QMP**
- The paper states that the proposed method outperforms QMP because "guide policy learns long-term policy guidance." I am not entirely convinced. One can simply use QMP to select from k-step behavior proposals and execute the selected proposal, which can also achieve this long-term policy guidance. Including this variant of QMP would be necessary to show that the performance gain comes from other designs of the proposed method.

**Guide step K**
- The best guide step K in the MT Half-Cheetah and Meta-World are different. It is unclear, given a new MTRL domain, how we should choose K. It seems that we could only tune this hyperparameter via trial and error.

**Backbone RL algorithms**
- This work adopts SAC as the backbone RL algorithm. Is it possible to use other RL algorithms, such as TD3 or PPO?

**Clarity**
- Section 4 is difficult to follow. Sufficiently describing the intuitions before introducing each component of the proposed method would significantly improve the readability of this section.

**Novelty**
- The proposed method seems incremental given that the QMP paper explores this idea of behavior sharing. Despite the high similarity between this work and the QMP paper, the authors seem to deliberately hide this by avoiding discussing QMP in the introduction.

**Questions:**

See above

**Limitations:**

Yes

---

> ### Author Rebuttal · Authors · 2024-08-06
>
> Thank you very much for your constructive feedback and acknowledgment of our efforts. Following are our responses to all your concerns.
>
> > Sample efficiency comparison.
>
> Please refer to **Figure 10 (Appendix D.1)**, which presents the full training curves for our main experiment (Table 1). It shows that beyond the ultimate performance improvement, CTPG also enhances the sample efficiency.
>
> > Comparison to K-step QMP.
>
> In our experiments, we use the hyperparameter settings from the original QMP paper. Thus, the QMP in our experiment is exactly the K-step QMP ($K$ = 10 in implementation) you asked for, which as you mentioned, shows that the performance gain comes from other designs of the proposed method. We are sorry for the ambiguity and we will add this hyperparameter setting to the Appendix in the revised manuscript. Thank you again for pointing this out.
>
> > Hyperparameter guide step $K$.
>
> Guide step $K$ is a necessary hyperparameter introduced by CTPG and may need slight adjustment across testbeds. However, we want to emphasize that the improvement gains of CTPG do not rely on the careful tuning of $K$ at all. As shown in **Fig.2 of the global Rebuttal Comment PDF** (a clear version of Figure 12), the performance improvement is evident even without tuning $K$. If the user pursues the potential best performance, we recommend exploring the value of $K$, and it can be selected easily with the help of hyperparameter search methods, such as Bayesian Optimization, Hyperband, etc. In the Conclusion and Discussion Section, we have already indicated that our future work will focus on dynamically selecting the guide step $K$.
>
> In addition, the output of CTPG's guide policy can add another action dimension to choose a specific guide step $K$. This approach not only automatically selects the $K$ value in different environments but also dynamically selects it at different time steps. We already have preliminary experimental results demonstrating the effectiveness of this scheme, and it also shows the high scalability of the CTPG framework. However, we believe that the contribution could be a new work, so it is mentioned only in future work.
>
> > Adaptability of other RL algorithms to CTPG.
>
> CTPG is a general MTRL framework that can be adapted with other RL algorithms, *even allowing the control and guide policy to use different RL algorithms*. Since SAC is a widely used algorithm in continuous control and serves as the base algorithm for all baselines, we also choose SAC as the base algorithm in this work. In addition, we explore the adaptation of TD3 with the CTPG and also employ different RL algorithms for the control and guide policies. Specifically, the control policy uses TD3, and the guide policy uses DQN. This approach is compared with the TD3-based MTRL method, and the result is displayed in **Fig.3 of the global Rebuttal Comment PDF**. The result shows that CTPG can enhance performance and sample efficiency when combined with other backbone RL algorithms.
>
> > Section 4 is difficult to follow.
>
> Thank you very much for your suggestion. We will include an additional intuition before introducing each module to enhance readability in the revised manuscript.
>
> > Similarity with QMP.
>
> We highly respect QMP and believe that QMP is an excellent work that greatly inspired our research. We discussed their differences in Related Work Section. Moreover, QMP serves as the most important and the only baseline in our experiments, and we openly compared our method against it. Following your suggestion, we will include a more detailed introduction of QMP and its great contribution in the revised Introduction.
>
> Here, we would like to re-emphasize the difference between CTPG and QMP: QMP uses the maximum Q-value of a single step to select the shared behavior over continuous K steps, which *only* guarantees the optimality of the first step of the shared policy. In contrast, CTPG learns a guide policy with the action space being exactly the task control policies to identify useful sharing policies for guidance by considering the benefits over K steps collectively. Additionally, CTPG proposes two gating mechanisms to avoid negative transfer from unhelpful policies, thereby improving sample efficiency.
>
> **Thank you once again for your valuable feedback. We hope our response has satisfactorily addressed your concerns. If you find that we have addressed your concerns, we kindly hope you reconsider your rating. If you need further elaboration or additional points to include in the response, we welcome further discussion to ensure everything is clear and satisfactory.**

---

> > ### Comment · Reviewer_5DoG · 2024-08-09
> > **Re: Rebuttal by Authors**
> >
> > Thank you for the rebuttal, including the additional TD3 results and clarifications. I am increasing my score to 5 to reflect what's addressed by the author's rebuttal.

---

> > > ### Author Response · Authors · 2024-08-10
> > >
> > > We sincerely thank you for taking the time to read our response. We are pleased that our response has effectively addressed your concerns. Thank you once again for your valuable input.
> > >
> > > Best regards,
> > >
> > > All authors

---

### Official Review · Reviewer_xTD7 · 2024-07-15

**Soundness:** 2
**Presentation:** 3
**Contribution:** 2
**Rating:** 7
**Confidence:** 3

**Summary:**

This paper proposes a method called CTPG to enable policies trained on different tasks to learn from each others' generated trajectories. CTPG operates by learning a guideline policy that determines which control policy in a given set should best generate trajectories to enable agents to learn a particular task. Then, the authors also designed additional mechanisms that (i) deal with how each policy whose trajectories are sampled may change from their learning process, (ii) prevent negative transfer from using trajectories from irrelevant policies for learning, and (iii) promote the convergence towards good policies by preventing any transfer of experience once a policy is already proficient at its intended task.

Using environments from multi-task reinforcement learning benchmarks, the authors then investigated their method's performance when (i) learning a set of tasks in parallel and learning a new task that is not from the set of previously learned tasks. Furthermore, the authors also conducted further analysis to investigate (i) which component within their learning algorithm is most responsible for the method's performance and (ii) whether the guideline policy learned a sensible experience selection policy.

**Strengths:**

**Major Strength - Clarity**

Except for a few minor clarification details provided below, I find the paper to be well-written. I especially appreciate the clear outlining of (i) the motivation behind the proposed method, (ii) the questions being investigated by the authors, and (iii) the experiment design. I hope that future iterations of the manuscript will keep this same level of clarity.

**Major Strength - Method Soundness**

From a high-level perspective, it seems that the learned transfer learning mechanism proposed in this paper seems reasonable. I especially find the authors' design of various mechanisms to address different problems (i.e., policy selection, negative transfer, variable rates of learning between tasks) to be great choices that should improve learning performance in multi-task reinforcement learning scenarios.

**Major Strength - Experiments and Analysis**

I also think that the authors did a great job of thoroughly investigating the efficacy of their method. The five questions in the experiment section provided interesting insights into the method. At the same time, the experiments done to answer each of those questions were well-designed and showed the method's positive performance.

**Minor Strength - Significance**

While allowing agents to learn from each others' experience to improve their performance at achieving their respective objectives is not exactly new, to my knowledge, the authors' proposed method for learning in multi-task reinforcement learning (and addressing its various associated issues) is novel. Even if I am wrong on the novelty aspect of this method, I still believe the thorough analysis provided by the authors would contribute to different insights that may be useful for the multi-task RL community.

**Weaknesses:**

**Minor Weaknesses - Additional Comparisons**

Perhaps another baseline that could be compared is the simple strategy from [1], where one simply learns from another policy with the highest performance at the task. Using this baseline should elucidate the effects of switching between different policies to transfer from as opposed to just transferring from a policy that seems to perform the best.

**Minor Weaknesses - Hindsight Off-Policy Correction**

From Section 4.1 alone, I also do not find how the hindsight off-policy correction mechanism affects the remainder of the learning process. While I did check Algorithm 2 in the appendix, I am uncertain why $j^{'}_{t}$ (the behavioral policy chosen in hindsight) only affects SAC's critic updates and not also the policy updates.

References:

[1] Disentangling Transfer in Continual Reinforcement Learning. Wolczyk et al. NeurIPS 2022.

**Questions:**

1. How would CTPG perform against a simple transfer learning strategy where experience is only transferred from the policy that has the best performance?

2. Does the behavioral policy chosen in hindsight affect the actor-network training? If it does, how would it do so?

**Limitations:**

The authors have sufficiently outlined their method's limitations and interesting directions for future work in the last paragraph of the manuscript.

---

> ### Author Rebuttal · Authors · 2024-08-06
>
> Thank you very much for your constructive feedback and acknowledgment of our efforts. Following are our responses to all your concerns.
>
> > Additional comparison with a simple transfer learning strategy where experience is only transferred from the policy that has the best performance.
>
> [1] focus on continual RL, requiring a predefined task learning sequence, which does not match the experimental setting of MTRL, so we cannot directly compare it with CTPG and other baselines. Therefore, we implement a comparable version based on the core idea of [1] and your description: selecting the policy that performs best in the current task among all task policies to guide exploration in generating trajectory data. Specifically, after each $H$ rounds of data collection, we evaluate all policies across all tasks and choose the best-performing policy for each task to collect the data for the next $H$ rounds. We refer to this baseline as BPT (Best Performance Transfer) and set $H$ to 50 episodes in implementation. We use MTSAC in HalfCheetah-MT8 and MHSAC in MetaWorld-MT10. The result, shown in **Fig.1 of the global Rebuttal Comment PDF**, demonstrates that CTPG outperforms BPT. CTPG's guide policy learns a more flexible strategy: it not only can learn to share a single policy within a complete trajectory (like BPT), but also can develop a mixture policy by combining different control policies, making it more transferable between tasks. Notably, BPT performs better in HalfCheetah-MT8 than in MetaWorld-MT10 because the tasks in HalfCheetah-MT8 share more significant similarities, whereas the policy transfer and guidance in MetaWorld-MT10 require combination strategies.
>
> [1] Disentangling Transfer in Continual Reinforcement Learning. Wolczyk et al. NeurIPS 2022.
>
> > The effect of the hindsight off-policy correction mechanism on SAC's actor network training.
>
> The hindsight off-policy correction mechanism affects not only the critic update in SAC but also the actor update.
> Although the actor loss (Line 4 in Algorithm 2) does not explicitly use relabeled $j'_t$, it indirectly affects actor learning due to SAC's unique actor update method [2]. Specifically, SAC's actor loss is derived from its optimization objective, and the actor policy is updated according to:
>
> $
> \pi_{new} = \arg\min_{\pi' \in \Pi} D_{KL} \left( \pi'(\cdot | s_t) \left\| \frac{\exp\left(\frac{1}{\alpha} Q^{\pi_{old}}(s_t, \cdot) \right)}{Z^{\pi_{old}}(s_t)} \right. \right),
> $
>
> where the partition function $Z^{\pi_{old}}(s_t)$ normalizes the distribution. In essence, SAC's policy is updated by fitting the distribution of the SoftMax function of Q-value with temperature $\alpha$. Therefore, modifying the update of $Q(j_t|s)$ to $Q(j'_t|s)$ using hindsight off-policy correction leads to a different actor optimization objective, thus affecting the training of the actor network. We will also elaborate on how the hindsight off-policy correction mechanism affects guide policy training in the revised manuscript.
>
> [2] Soft Actor-Critic: Off-Policy Maximum Entropy Deep Reinforcement Learning with a Stochastic Actor. Haarnoja, Tuomas, et al. ICML 2018.
>
> **Thank you once again for your valuable feedback. We hope our response has satisfactorily addressed your concerns. If you need further elaboration or additional points to include in the response, we welcome further discussion to ensure everything is clear and satisfactory.**

---

> ### Comment · Reviewer_xTD7 · 2024-08-11
> **Official Comment from Reviewer xTD7**
>
> Thank you for considering attempting to address the points I've previously raised.
>
> > Specifically, after each $H$ round of data collection, we evaluate all policies across all tasks and choose the best-performing policy to collect the data for the next $H$ rounds. We refer to this baseline as BPT (Best Performance Transfer)
>
> This is exactly the baseline that I envisioned in my previous feedback. Given the positive results when comparing against this baseline, I no longer view the lack of comparisons against BPT as a minor weakness of this paper.
>
> > Although the actor loss (Line 4 in Algorithm 2) does not explicitly use relabeled $j'_t$, it indirectly affects actor learning due to SAC's unique actor update method [2].
>
> I also agree that this needs to be better highlighted in the next manuscript version. Perhaps it's also important to note that unlike SAC-based actor updates in environments with continuous action spaces (where we rely on a Monte Carlo estimate of the KLD objective since integrating over each possible action is impossible), the KLD objective can be easily evaluated when the input distributions to the KL divergence are categorical.
>
> **Closing remarks**
>
> Given the authors' responses to the points raised by each reviewer, I am increasing my score because I believe this is a good paper worthy of acceptance.

---

> > ### Author Response · Authors · 2024-08-11
> >
> > Thank you very much for taking the time to review our response. We are glad that our response has effectively addressed your concerns. Once again, we sincerely appreciate your thorough review and insightful feedback, which have helped us enhance our paper.
> >
> > Best regards,
> >
> > All authors

---

### Author Rebuttal · Authors · 2024-08-06

To AC and all the reviewers:

We would like to express our sincere gratitude to AC and all the reviewers for their great efforts in evaluating our paper. Your valuable insights and suggestions are greatly appreciated. We have carefully addressed all the questions and concerns raised in the reviews through our rebuttal comments.

The attached PDF to this global rebuttal comment contains:

* Figure 1 is the experimental results with an additional baseline BPT (Best Performance Transfer).

* Figure 2 is an extended version of the ablation study on guide step $K$ (Figure 12 in paper), which includes an additional curve base MHSAC for convenient comparison.

* Figure 3 is the additional experimental results of adapting CTPG to the base RL algorithm TD3.

* Figure 4 is the experiment results on MetaWorld-MT10 with x-axis being wall-clock time.

If there are any remaining queries or uncertainties after reviewing our responses, we welcome further discussions during the upcoming phase. Your continued engagement is highly valued and appreciated. Thank you once again for your time, expertise, and contribution to our paper.

---

### Comment · Area_Chair_WDMG · 2024-08-07

Dear Reviewers,

The author responses have been uploaded. Please carefully review these responses to see if your concerns have been adequately addressed and actively participate in discussions.

It is important that **all reviewers should acknowledge having read the author responses by posting a comment**, irrespective of whether there is any change in your rating.

Thank you for your cooperation.

Best regards, \
Area Chair

---

### Author Response · Authors · 2024-08-14
**Rebuttal Summary and Appreciation**

Dear Reviewers, ACs, SACs, and PCs:

As the discussion phase comes to a close, we would like to sincerely express our gratitude for your time and effort throughout the review and rebuttal process. We deeply appreciate the constructive and insightful comments provided by the reviewers after carefully reading our manuscript, which have significantly improved the quality of our work.

Here, we would like to summarize the key points of our rebuttal and discussions:

1. Based on (Reviewer xTD7)'s feedback, we conducted ***additional experiments comparing CTPG with BPT***. These experiments more clearly demonstrate the advantages of CTPG in learning a mixture policy by combining different control policies.
2. In response to (Reviewer 5DoG)’s concerns, we performed ***supplementary experiments using another base RL algorithm TD3***, which further demonstrates that CTPG is a generalized MTRL framework.
3. To address (Reviewers J1dv and k7nL)’s concerns, we provided a complementary ***analysis of the computational complexity*** of CTPG.
4. We are grateful for all the reviewers’ comments on the revisions, which have made our paper more complete, including ***a detailed explanation of the hindsight mechanism's impact on SAC*** (xTD7), ***additions to the experimental hyperparameters setup*** (5DoG), ***a more relevant realistic example with CTPG*** (J1dv), and corrections of some typos.

***During the discussion, all the reviewers provide appreciations and claims for his/her concerns addressed,*** and we were ***especially encouraged*** by (Reviewer xTD7)’s conclusion: ***"Given the authors' responses to the points raised by each reviewer, I am increasing my score because I believe this is a good paper worthy of acceptance."***. We appreciate that we receive ***all positive*** feedbacks from every reviewer. We believe that our work would offer valuable insights to the MTRL community and make a positive contribution to the `NeurIPS 2024` conference.

We would like to thank you once again for your time and dedication throughout the review process.

Best regards,

Authors

---

### Decision · Program_Chairs · 2024-09-25

**Decision:**

Accept (poster)

**Comment:**

The paper introduces Cross-Task Policy Guidance (CTPG), a method for enhancing multi-task reinforcement learning (MTRL) by allowing policies to learn from each other's trajectories. The approach involves a guide policy that selects the most suitable behavior policy from a pool of all tasks' control policies, along with mechanisms to prevent negative transfer and ensure efficient learning. CTPG shows improved performance in benchmarks like MetaWorld and Half-Cheetah.

While the method is innovative, some reviewers noted that the contribution feels incremental, especially in comparison to latest methods such as QMP. Concerns were also raised about the scalability of CTPG to more complex tasks and the focus on converged performance rather than sample efficiency. The computational complexity of the approach was another point of uncertainty.

However, the paper also exhibits strengths. The paper is well-presented, with clear motivation and thorough experiments that demonstrate the effectiveness of CTPG. The innovative approach to leveraging cross-task similarities offers potential for advancing MTRL research. The positive impact and the detailed response to reviewer concerns support its acceptance.

Based on the overall positive reviews, it is recommended to accept the paper, provided that the revisions discussed in the rebuttal and comparison to the most recent multi-task RL algorithms are incorporated in the final version.